# Unlocking the Potential of *Ganoderma lucidum* (Curtis): Botanical Overview, Therapeutic Applications, and Nanotechnological Advances

**DOI:** 10.3390/pharmaceutics17040422

**Published:** 2025-03-26

**Authors:** Ana Eira, Maria Beatriz S. Gonçalves, Yannick Stéphane Fotsing Fongang, Cátia Domingues, Ivana Jarak, Filipa Mascarenhas-Melo, Ana Figueiras

**Affiliations:** 1Laboratory of Drug Development and Technologies, Faculty of Pharmacy, University of Coimbra, 3000-548 Coimbra, Portugal; anaeiraa@gmail.com (A.E.); mariabsgoncalves@gmail.com (M.B.S.G.); cdomingues@ff.uc.pt (C.D.); jarak.ivana@gmail.com (I.J.); 2Higher Teachers’ Training College, The University of Maroua, Maroua P.O. Box 55, Cameroon; fongangfys@yahoo.fr; 3Associated Laboratory for Green Chemistry (LAQV) of the Network of Chemistry and Technology (REQUIMTE), Group of Pharmaceutical Technology, University of Coimbra, 3000-548 Coimbra, Portugal; 4Institute for Clinical and Biomedical Research (iCBR) Area of Environment Genetics and Oncobiology (CI MAGO), Faculty of Medicine, University of Coimbra, 3000-548 Coimbra, Portugal; 5Higher School of Health, Polytechnic Institute of Guarda, 6300-307 Guarda, Portugal; 6BRIDGES—Biotechnology Research, Innovation and Design of Health Products, Polytechnic University of Guarda, 6300-559 Guarda, Portugal

**Keywords:** *Ganoderma lucidum*, nanotechnology, nanoparticles, triterpenoids, polysaccharides, bioactive compounds, drug delivery, cancer therapy

## Abstract

**Background:** *Ganoderma lucidum* (*GL*), commonly known as the “Lingzhi” or “Reishi” mushroom, has long been recognized for its potential health benefits and medicinal properties in traditional Chinese medicine. The unique potential combination of bioactive compounds present in *GL*, such as triterpenoids, polysaccharides, and peptides, has inspired interest in leveraging their therapeutic potential In recent years, the emerging field of nanotechnology has opened up new possibilities for using the remarkable properties of *GL* at the nanoscale. **Objetive:** The main objective of this review is to explore the unique potential of *GL* in traditional and innovative therapies, particularly in cancer treatment, and to assess how nanotechnology-based strategies can enhance its therapeutic applications.is to explore. **Results:** Nanotechnology-based strategies have been investigated for the efficient extraction and purification of bioactive compounds from *GL*. Additionally, nanocarriers and nanoformulations have been developed to protect these sensitive bioactive compounds from degradation, ensuring their stability during storage and transportation. The use of *GL*-based nanomaterials has shown promising results in several biomedical applications, namely due to their anticancer activity by targeting cancer cells, inducing apoptosis, and inhibiting tumor growth. **Conclusions:** The combination of *GL* and nanotechnology presents an exciting frontier in the development of novel therapeutic and biomedical applications. Nevertheless, further research and development in this interdisciplinary field are warranted to fully exploit the synergistic benefits offered by *GL* and nanotechnology. Future prospects include the development of robust clinical trials focused on *GL* nanotechnology-based cancer therapies to clarify mechanisms of actions and optimize formulations, ultimately leading to innovative solutions for human health and well-being.

## 1. Introduction

There is a growing interest in exploring bioactive ingredients and their therapeutic activities in cancer treatment.

In 2022, the World Health Organization (WHO) and International Agency for Research on Cancer (IARC) reported 20 million new cancer cases and 9.7 million deaths worldwide, with 53.5 million people living five years after diagnosis. The data highlight that 1 in 5 people will develop cancer, and 1 in 9 men and 1 in 12 women will die from it. This underscores the urgent need to address cancer inequities and improve care [1].

The global awareness of functional foods has significantly expanded. Mushrooms have been used for health promotion and disease treatment since ancient times in Asia, whereas this practice has only gained widespread recognition in the Western countries more recently. Different mushroom species display a range of functions, owing to the distinct bioactive compounds they contain. Studies have highlighted that mushrooms exhibit a variety of complex biological activities, including moderate to strong antioxidant properties, as well as notable effects on immune system modulation. Although approximately 1.4 million mushroom species exist worldwide, only around 10% have been fully identified along with their species and metabolites [2,3].

In this regard, *GL*-derived compounds, such as triterpenoids and polysaccharides, have demonstrated activity through several mechanisms. Stimulation of the immune system, anti-proliferative properties, apoptosis of tumor cells, and cytotoxic effects are mentioned as some of the anticancer activities of these bioactive ingredients [4,5].

Cancer is characterized by the rapid proliferation of abnormal cells, usually originating from a single abnormal cell, highlighting the need for effective therapeutic strategies. These cells have lost their normal control mechanisms, enabling them to continuously multiply. As a result, they surpass their typical boundaries, invade neighboring tissues, and have the potential to migrate to distant parts of the body, spreading to other organs [4,6]. The formation of cancer occurs through the transformation of normal cells into tumor cells, following a multistage process that generally progresses from a pre-cancerous lesion to a malignant tumor. In this regard, symptoms can vary significantly based on the cancer’s location, type, size, and extent [7,8,9]. Although some slow-growing cancers may not necessarily require treatment, the vast majority require active treatment. An accurate cancer diagnosis is essential for ensuring appropriate and effective treatment.

While chemotherapy plays a major role in early and second stages, it may not work properly if it faces resilient cells, when the tumor starts to regrow. In addition, treatments are continuously improving to reduce the side effects. Therefore, the scientific community focuses on finding therapeutic alternatives to overcome these challenges, such as drug targeted delivery [9].

Nanotechnology offers unique opportunities to address limitations and enhance the therapeutic efficacy of *GL* in cancer treatment. The use of nanoscale materials, such as nanoparticles, allows for the encapsulation and delivery of *GL* bioactive compounds, providing several advantages. Nanoparticles can protect the compounds from degradation, improving their solubility, and enable controlled release, thereby enhancing their bioavailability, and extending their circulation time in the body [5]. Furthermore, nanotechnology enables the development of targeted drug delivery systems using *GL*-based nanocarriers [5]. The functionalization of nanoparticles with targeting ligands, such as antibodies or peptides, can enhance their specific accumulation at the tumor site. This approach minimizes off-target effects and improves therapeutic outcomes [10]. Additionally, nanotechnology-based strategies can overcome multidrug resistance, a major challenge in cancer therapy, by co-delivering *GL* compounds with chemotherapeutic agents or by modulating drug efflux mechanisms [11]. In summary, the combination of *GL* and nanotechnology holds immense promise for advancing cancer therapy. The integration of nanomaterials with *GL* also offers opportunities for cancer diagnostics and imaging [12]. Further research and development in this interdisciplinary field are crucial to harness the full potential of *GL* in nanotechnology-based cancer therapy, ultimately leading to improved treatment outcomes and patient well-being.

Implementing strategies to enhance quality and safety control procedures is imperative to establish clear standards and consistency for *GL* nanotechnology formulations. These measures are essential to gain a comprehensive understanding of the mechanisms through which these formulations exert their effects in cancer therapy and to facilitate the characterization of their active components.

The primary goal of this review is to explore and leverage the synergistic potential of *GL* and nanotechnology to revolutionize cancer therapy, by providing an updated literature analysis on its pharmacological properties, toxicity profile, application in cancer therapy, preclinical and clinical trials, as well as regulatory considerations related to the use of new drug delivery nanosystems using *GL.* Nanoparticles are tailored for specific applications based on their unique properties. Metallic (gold or silver) and ceramic nanoparticles are favored for drug delivery. On the other hand, polymeric nanoparticles, derived from synthetic or natural polymers, are ideal for controlled drug release, improving bioavailability and reducing side effects. Lipid nanoparticles (LNPs), like solid lipid nanoparticles (SLNs) and liposomes, effectively contribute to delivering both hydrophobic and hydrophilic drugs with stability and controlled release. Carbon-based nanoparticles, quantum dots, and composite nanoparticles have also demonstrated interesting properties, making them suitable for complex medical and industrial uses [13].

This review will primarily focus on exploring the anticancer properties of *GL* polysaccharides, and triterpenes, along with their underlying mechanisms. However, the clinical application of *GL* active constituents may face some challenges. As bioactive compounds, it is important to ensure their stability and protect them from degradation [4,5]. In this regard, nanomaterials and innovative strategies aim to maximize the therapeutic potential of GL bioactive compounds, contributing to more effective treatments and advancing personalized approaches for cancer care.

## 2. Review Methodology

This review offers a comprehensive examination of the chemical composition, synergistic potential, and biological properties of natural products derived from the *GL* mushroom. Additionally, it explores the traditional medicinal uses of these products. The search terms included the keywords of this manuscript. The search was conducted using various electronic databases, including SciFinder, Pubmed, Science Direct, Dictionary of Natural Products, Web of Science, and Google Scholar. Other resources such as books, theses, and library materials were also utilized. Some of the following index terms were included: “advanced therapies medicinal products”, “nanotechnology”, “nanoparticles”, “nanomedicine”, “neoplasms”, cell therapy”, “gene therapy”, “drug formulation”, “triterpenoids”, “polysaccharides”, “bioactive compounds”, “drug delivery”, “cancer therapy”, among others. Particularly, the following MeSH terms were added: Nanoparticles; Nanotechnology; and *Ganoderma lucidum*. Other core databases were assessed, including ema.europa.eu/en, fda.gov, clinicaltrials.gov, clinicaltrialsregister.eu, and nih.gov, among others. When appropriate, the boolean operators “AND”, “OR” or “NOT” were applied. The inclusion criteria are the following: articles that contain one of the considered index terms in the title or in the abstract and are presented preferentially in English. At least one author read the title and the abstract of the manuscripts to select the articles to be included as bibliographic support in this work. Data organization was carried out using Microsoft Office 2016 software, and structural representations were created with ChemDraw Professional 16.0.

## 3. *Ganoderma lucidum*: Botanical Overview, Characterization, Uses in Traditional Medicine, and Chemical Studies

### 3.1. Botanical Overview and Characterization

The term *Ganoderma* derived from the Greek word «ganos», which refers to shinning, and «derma» for skin was established as a genus in 1881 by the Finnish mycologist Petter Adolf Karsten, initially encompassing only one species [14]. It was later revised by Patrouillard in his monograph (1889), where all species with pigmented spores, adhering tubes, and laccate crusted pilei were included, resulting in the recognition of 48 species [15,16]. The difficulty in classification arose from the lack of reliable morphological characteristics, the abundance of synonyms, and the widespread misuse of names.

Phylogenic analysis based on DNA sequence information has played a crucial role in clarifying the relationships among *Ganoderma* species. The genus *Ganoderma* (Ganodermataceae family) can now be categorized into six monophyletic groups with approximately 130 species of polypore wood-decaying fungi that can be annuals or perennials widely distributed in tropical regions [17,18].

*Ganoderma lucidum* is characterized by its sizeable and dark mushroom with a glossy exterior, exhibiting a woody texture, and grows on plum trees in Asia, also known as “Ling-zhi (meaning «spiritual power grass» in China and Korea, or “Reishi” or “Mannentake” in Japan), is a well-known medicinal fungus with a rich history of use in traditional Chinese medicine [19]. *GL* grows globally, thriving in temperate and subtropical areas of Africa, America, Canada, China, Europe, India, Japan, Korea, and other Southeast Asian countries [20].

Over the years, extracts of *GL* have been transformed into various forms, including tea, dietary supplements, and powder. These products are readily available in the market and are utilized for the treatment of different illnesses [21]. The physical properties can be characterized by assessing its appearance, texture, color, and moisture content. These properties can vary depending on the growth conditions, cultivation methods, and post-harvest processing [22].

Characterization studies on *GL* are ongoing and aim to provide a better understanding of its chemical composition, biological activities, and potential therapeutic applications. These studies contribute to the development of standardized extracts, formulations, and quality control measures for the safe and effective use of *GL* in various healthcare products. Characterization of *GL* involves analyzing its chemical composition, identifying and quantifying the active compounds, and studying its physical and biological properties [23].

*GL* has a distinctive appearance that sets it apart from other mushrooms (Figure 1). It typically has a large, flat, and kidney-shaped cap that can range in size from 5 to 25 cm in diameter [24]. The cap is smooth and shiny, with a reddish-brown color, but it can range from a lighter, rusty brown to a dark, almost blackish-brown color. The undersurface of the cap is usually white or light brown and may exhibit small, round pores [25]. The texture of the *GL* mushroom can vary depending on its age and growth conditions. When young, the cap is often soft and fleshy, but as it matures, it becomes harder and more woody [25]. The flesh of the mushroom is corky and tough, making it unsuitable for direct consumption. The color of *GL* can vary depending on its specific variety and growing conditions. *GL* mushrooms contain a significant amount of moisture when they are fresh and just harvested. However, during the drying process, the moisture content is reduced to increase their shelf life and facilitate storage. The exact moisture content can vary depending on the drying method used, but typically dried *GL* mushrooms exhibit a moisture content of around 10% or lower [22,23,24,25].

As mentioned, the colors of *GL* can vary depending on its specific variety, growing conditions, age, and other environmental factors. While *GL* is most commonly known for its red, black, and purple varieties, the colors can show variations due to the following factors:(1)Variety or Strain

Different varieties or strains of *GL* may exhibit distinct colors. Red, black, and purple are some of the most recognized color variations, but there may be other rare colors in specific strains, such as blue, yellow, or white [23,24,25].

(2)Growing Substrate

The substrate on which *GL* is cultivated or naturally grows can influence its color. Different substrates, such as different types of wood or other materials, may lead to variations in the color of the mushroom [23]. Temperature and humidity are crucial in determining the yield of bioactive compounds [26].

(3)Environmental Conditions

Environmental factors, including temperature, humidity, light exposure, and nutrient availability, can impact the pigmentation and color expression of *GL* [23]. Controlled environments can enhance the production of these compounds, influencing the mushroom’s medicinal properties [26].

(4)Age and Maturity

The age of *GL* affects its bioactive compound content, with mature fruiting bodies generally exhibiting higher concentrations of therapeutic compounds like triterpenoids and polysaccharides. This can also affect its color, since younger mushrooms may display different colors compared to mature ones [23,26].

(5)Genetic Expression

The expression of genes responsible for pigment production in *GL* can contribute to the observed color variations [27]. Genetic expression varies by strain, with genetically engineered strains capable of producing higher levels of these beneficial compounds [26].

Some color variations may be more common in certain regions or strains of GL, and color alone is not always a definitive indicator of the specific variety or quality of the mushroom [23]. Other factors, such as chemical composition and morphological characteristics, are also essential for proper identification and assessment.

### 3.2. Uses in Traditional Medicine

The *GL* mushroom has long been recognized in traditional practices, such as Traditional Chinese Medicine, and Japanese and Korean Traditional Medicine, as a “tonic for promoting longevity” and well-being, and has gained recognition as a valuable medicinal resource in the healthcare system [28]. In traditional medicine, it is commonly used through decoction [29]. As a ‘potent elixir” in Oriental culture, it is valued for its potential health benefits [30]. In China, it is linked to longevity, spiritual power, and overall well-being. Traditionally, it has been utilized in China to restore vital energy, promote relaxation of the mind, alleviate coughing, and relieve symptoms of asthma [19].

*GL* has historically served as a traditional remedy for conditions such as breathlessness, palpitations, dizziness, and insomnia [19]. Additionally, it has been recognized and extensively utilized as an adjuvant in the treatment of different cancer types including breast cancer [31,32].

**Figure 1 pharmaceutics-17-00422-f001:**
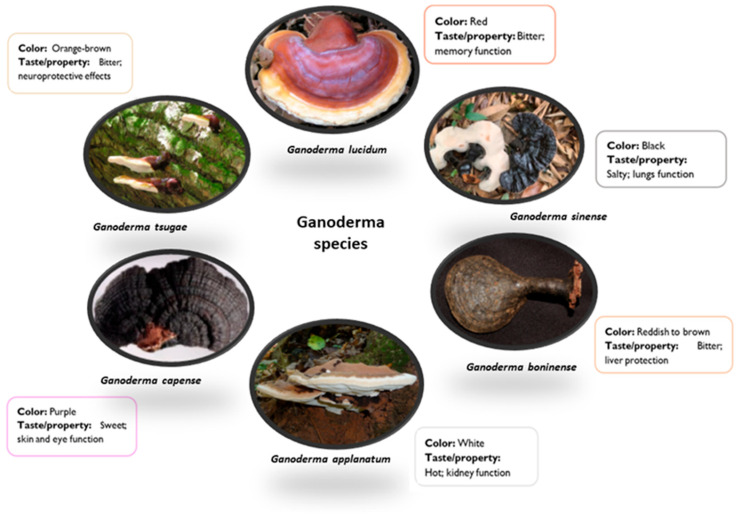
Principal *Ganoderma* species. Images adapted from [33,34,35,36,37,38].

### 3.3. Chemical Studies

Extensive research has been conducted on the *GL* mushroom, exploring its diverse bioactive compounds and medicinal properties. The huge market attention for *GL* mushroom is attributable to the wide range of bioactive compounds that it presents. *GL* contains an intricate combination of bioactive compounds, including polysaccharides, triterpenoids, proteins, peptides, nucleotides, sterols, and phenolic compounds. Characterization involves identification and quantification of these compounds using techniques such as chromatography (e.g., high-performance liquid chromatography, gas chromatography) and spectroscopy (e.g., mass spectrometry, nuclear magnetic resonance) [39].

About 400 bioactive compounds have been identified from various parts, including the fruiting bodies, mycelia, and spores of *GL* [23]. The *GL* mushroom’s compounds have been recently recognized to be a traditional optimal source of natural bioactive components, including alkaloids, polyphenols, polysaccharides (*α*/*β*-D-glucans), steroids, triterpenoids (ganoderic acids, ganoderenic acids, ganoderol, ganoderiol, lucidenic acids), nucleotides (5’—guanosine monophosphate and 5’—xanthosine monophosphate were found in both young and mature forms of the mushroom, although in a higher proportion in matured basidiocarps), nucleosides (adenosine, iosine, uridine), amino acids, minerals, trace elements, vitamins, and proteins [23,40].

To obtain and isolate the specific bioactive compounds from the mushroom extract, we have distinct steps, which include extraction, separation, and purification. The bioactivity of compounds from *GL* is mainly related to their molecular structure, not their ionization state. While ionization can affect properties like solubility, it does not usually impact their therapeutic effects. Therefore, the purification processes used to isolate these compounds are unlikely to alter their effectiveness [41].

In summary, extraction is the process of obtaining the bioactive compounds from *GL* using a solvent. Separation involves separating different compounds present in the extract, and purification further refines the compounds to obtain highly pure and concentrated forms of the bioactive compounds (Table 1).

It is important to note that the biological activities of GLPs can vary depending on factors such as extraction methods, molecular weight, and structural composition [56,57,58]. Typically, GLPs are extracted from the mushroom using hot water and subsequently precipitated with ethanol or methanol [41]. Because GLPs are prone to oxidative deactivation, it is crucial to employ a suitable encapsulation method to prevent oxidation [57]. Additionally, sensitive materials can be encapsulated within the matrix, providing protection for the bioactive compounds against the external environment, which is advantageous for GLPs [59]. Further purification of the extracts can be achieved through various separation methods, including both normal and reverse-phase High-Performance Liquid Chromatography (HPLC) [41]. According to the literature, recent studies performed on *GL* have shown that both its total phenolic content (TPC) and total flavonoid content (TFC) vary depending on growth stages and cultivation conditions. Research indicates that mature fruiting bodies tend to have higher levels of phenolics and flavonoids compared to younger stages. Flavonoid levels are also affected by factors such as the solvent used during extraction, with ethyl acetate extracts showing higher TFC. These variations highlight the importance of optimizing cultivation and extraction processes to maximize the therapeutic potential of *GL* [60,61].

In this regard, fractionation studies on *GL* focus on isolating bioactive compounds, primarily triterpenoids, which are key to its medicinal properties. Techniques like liquid–liquid extraction and chromatographic methods have been used to enrich and separate these metabolites. Recent advancements include the preparation of triterpenoid-enriched fractions (TEFs) and their pharmacokinetic profiling, revealing rapid absorption and bioavailability [62]. In addition, another study discusses the synergy between asymmetrical flow field-flow fractionation (AF4), ultrafiltration, and a deproteinization technique. This approach effectively removes protein impurities while preserving their structural integrity [63].

According to the literature, a recent study assessed the anticancer potential of extracts from fresh *GL* mushrooms. These extracts demonstrated significant activity against various cancer cell lines, including breast and colorectal cancers, with effective concentrations inhibiting growth. The analysis identified 13 triterpenoids and 28 phenolic compounds, highlighting ganoderic acid derivatives as the predominant triterpenoids and resveratrol as the most abundant phenolic. These compounds contribute to the observed effects, proving their potential as a source of anticancer bioactive compounds [64].

## 4. Pharmacological and Toxicological Properties

### 4.1. Pharmacological Properties

To date, approximately 279 secondary metabolites have been isolated from different parts of the mushroom, including the fruit bodies, mycelium, and spores [23]. While research on *GL* and its bioactive compounds is ongoing, some of the pharmacological properties reported are immunomodulatory activity (*GL* is known for its immunomodulatory effects, which means it can modulate the immune system) [23,42,43,56,65], antioxidant activity [23,42,43], anti-inflammatory activity [23,42,43,56], and anticancer potential [42,56,65]. It is also known for its role in regulating blood pressure, reducing cholesterol levels, inhibiting platelet aggregation, and improving blood flow [23]. In addition, antidiabetic, antiviral, and neuroprotective effects have also been described [56,65,66,67,68]. Regarding antiviral activity, *GL* has demonstrated significant potential in COVID-19 treatment. It has been shown to reduce the harmful effects of the virus on hematological and immunological responses. Additionally, *GL* exhibits dose-dependent inhibition of SARS-CoV-2 enzymes, highlighting its antiviral capabilities. These findings suggest that *GL* could be a valuable source of natural compounds with anti-coronavirus activity [69,70,71]. The pharmacological activities of these bioactive compounds are attributed to their interactions with various cellular targets and signaling pathways. It is crucial to highlight that understanding the pharmacology of the bioactive compounds in *GL* is a complex task. Further research is necessary to comprehensively uncover their mechanisms of action and therapeutic potential. While polysaccharides and triterpenes have been highlighted as the main components, *GL* also contains other bioactive compounds, including adenosine [23]. These compounds have been associated with interesting activities, including antiviral, anti-inflammatory, and hepatoprotective activities [65]. Figure 2 describes the correlation between *GL* compounds and their pharmacological properties.

#### 4.1.1. Polysaccharides

*GL* polysaccharides (GLPs) represent the primary category of bioactive compounds identified in *GL*. These complex carbohydrates exhibit immunomodulatory, antitumor, and antioxidant properties [23,56,65]. They can stimulate immune cells, such as macrophages, natural killer cells, and T cells, thereby promoting heightened immune responses against cancer cells [56,66]. Additionally, GLPs have shown potential in modulating inflammatory processes, and promoting the body’s defense mechanisms [59,66]. Polysaccharides make up 10–50% of the dry matter of fruiting bodies [57].

A comprehensive search has revealed over 200 different polysaccharides from spores, fruiting bodies, and mycelia, including *β*-D-glucans, *α*-D-glucans, *α*-D-mannans, and polysaccharide–protein complexes [56,57]. The immunomodulatory effects of GLPs are credited to their capacity to interact with immune receptors and signaling pathways. In addition to their immunomodulatory properties, GLPs also possess potent antioxidant activity. By reducing oxidative damage, these polysaccharides may contribute to overall health and well-being. Furthermore, studies have indicated that GLPs exhibit potential anticancer effects. They have been shown to inhibit the growth and proliferation of cancer cells, induce apoptosis (programmed cell death), and inhibit angiogenesis (the formation of new blood vessels that supply tumors) [58].

Among these bioactive compounds, β-glucans play a significant role in various diseases due to their pharmacological properties [56]. *β*-Glucans derived from *GL* have demonstrated immunomodulatory and anticarcinogenic features. Their biological activity is influenced by several factors, including molecular size, branching, water solubility, and overall form. The presence and arrangement of lateral branches, as well as the length of lateral chains, can impact their pharmacological properties. The ratio of the number of bonds within *β*-glucans also affects their activity [56,57,59]. Regarding their immunomodulatory properties, they have been shown to enhance the immune system’s response by activating immune cells such as macrophages, natural killer cells, and dendritic cells [56,57,58]. This effect is beneficial in various diseases and conditions where immune function plays a crucial role.

#### 4.1.2. Triterpenes

Numerous highly oxygenated and pharmacologically active triterpenes, including ganoderic acids, ganoderiol, lucidenic acid, lucialdehyde, ganolucidic acids, lanostanoid, ganodermatriol, and ganodermanontriol, have also been identified in *GL* [59]. Compounds that have a carboxyl group and fall under the category of triterpenoids are commonly referred to as ganoderic acids. These substances are identified by their intricate structure, substantial molecular weight, and notable lipophilicity. They represent highly oxidized derivatives of lanostane and their molecular structure may consist of 30, 27, or 24 carbon atoms [56,57,59]. The chemical structure of the main GL compounds is illustrated in Figure 3.

Alongside polysaccharides, triterpenoids showcase diverse pharmacological activities, encompassing anti-inflammatory, antioxidant, and hepatoprotective effects [23,59,66]. They have also shown potential for antitumor activity by inhibiting cancer cell growth, inducing apoptosis, and suppressing angiogenesis [59]. Recent research on *GL* has underscored its anticancer potential, especially through bioactive compounds like polysaccharides and triterpenoids. These substances are involved in multiple biological processes, such as hindering cell growth and modulating the body’s defense mechanisms. The required concentration to inhibit cell proliferation can differ depending on the compound; polysaccharides may show efficacy at lower concentrations, while triterpenoids could need higher levels to produce similar outcomes. Additionally, factors like formulation type and cancer variety affect the required dosage [43].

### 4.2. Toxicological Properties

*GL* is generally regarded as safe for consumption and boasts a lengthy history of use in traditional medicine. In this regard, it is generally recognized as safe (GRAS) and commonly consumed as a food supplement with minimal reports of acute toxicity. However, it is important to consider the potential toxicological properties and safety aspects associated with any substance, including *GL*.

Although rare, allergic reactions, such as skin rashes or respiratory symptoms, may occur, particularly in those with mushroom or fungal allergies. *GL* may interact with medications by inhibiting liver enzymes, affecting drug metabolism. Product quality can vary, with risks of contamination by heavy metals or pesticides, emphasizing the need for reliable sources. Pregnant or lactating women, infants, and individuals with certain health conditions should exercise caution [23]. These potential adverse effects and drug interactions are summarized in Table 2.

Recent in vivo toxicological studies have evaluated *GL* polysaccharide-based nanoparticles as drug delivery systems in animal models. These nanoparticles were used to deliver doxorubicin, which resulted in improved immune responses and tumor suppression in mice without significant toxicity. They also enhanced macrophage activation and CD8+ T cell responses, promoting effective cancer treatment while minimizing side effects. Another study confirmed that the nanoparticle formulation reduced systemic toxicity compared to traditional drug delivery methods, demonstrating its potential for safer, targeted cancer therapies in animal models [84].

Overall, while *GL* shows promise as a therapeutic agent, it is crucial to be aware of the potential toxic effects, allergic reactions, and interactions with certain medications.

### 4.3. Dosage Forms and Posology

*GL* can be prepared in various dosage forms and consumed on different routes. The appropriate dosage and posology of *GL* may depend on factors such as the specific formulation, the individual’s age, overall health, and the intended purpose of use. Some of the common dosage forms and posology options for *GL* already present in the market are as follows:***Capsules or Tablets***: *GL* is commonly available in the form of capsules or tablets. The recommended dosage may vary depending on the concentration of *GL* extract or powder in each capsule/tablet. A common dosage range is 1–3 capsules/tablets per day, taken with water or as directed by a healthcare professional [21,83].***Powder***: *GL* powder can be mixed with water, juice, smoothies, or other beverages. Dosage may fluctuate depending on the particular product and the desired effects. Generally, a typical dosage range is 1–3 g of *GL* powder per day [21,83]. It is advisable to start with a lower dosage, and gradually increase if needed, based on individual tolerance and response.***Extracts***: *GL* extracts are available in liquid or concentrated forms [21,83]. These extracts are often standardized to contain specific amounts of bioactive compounds. The dosage and posology for *GL* extracts may vary based on the concentration and potency of the extract.***Tea or Decoction***: *GL* can be brewed as a tea or decoction. Dried *GL* slices or powder can be simmered in water for a certain period to extract the bioactive compounds. The dosage of *GL* tea or decoction can vary depending on the concentration, brewing time, and individual preferences [19,21]. It is recommended to start with a small amount, and adjust the dosage based on taste and individual response.***Topical Formulations***: *GL* extracts or creams are also available for topical use. These formulations are often used for skin health or cosmetic purposes [21,83]. The dosage and posology for topical products may depend on the specific formulation and intended use (Table 3).

To date, these formulations are marketed, mostly, as dietary supplements. Companies such as Mountain Rose Herbs, BulkSupplements, and Bio-Botanica provide GL extracts and powders, while others such as Foodicine focus on tablets and capsules. Topical *GL*-based formulations are also commercialized, for example, by TRI-K Industries [85,86,87,88,89].

The optimal dosage form and posology of *GL* can differ based on individual factors and specific health goals. Consulting with a healthcare professional is recommended to determine the appropriate dosage and posology for your specific needs, and to ensure safe and effective use of *GL*.

## 5. Application of *Ganoderma lucidum* in Cancer Therapy

Cancer is a complex and devastating condition marked by uncontrollable cell proliferation and the capacity to metastasize. Conventional cancer treatments, like chemotherapy, radiation therapy, and surgery, have improved patient outcomes; however, they often come with significant side effects and limitations. *GL* encompasses bioactive substances, including polysaccharides, triterpenes, and other phytochemicals, that have been researched for their potential anticancer attributes. These compounds may exert various effects on cancer cells and the immune system, making them attractive candidates for complementary or alternative approaches in cancer therapy [90]. *GL* affects multiple genes with a major role in cancer treatment. It reduces metastatic potential by suppressing the urokinase-type plasminogen activator (uPA) and its receptor, urokinase plasminogen activator receptor (uPAR), which are involved in tumor cell invasion. In addition, there is an increasing expression of immune-regulating genes, boosting cytokines such as tumor necrosis factor-alpha (TNF-α), interferon (IFN), and interleukins, which activate macrophages and T-helper cells [91].

The potential mechanisms by which GL may impact cancer therapy include the following [32]:***Inducing Apoptosis***: *GL* may promote apoptosis (programmed cell death) in cancer cells, inhibiting their uncontrolled growth and survival.***Modulating the Immune System***: *GL* is known for its immunomodulatory effects, enhancing the activity of immune cells pivotal in identifying and eradicating cancer cells.***Reducing Inflammation***: Chronic inflammation has been linked to cancer development and progression. *GL*’s anti-inflammatory properties may contribute to controlling tumor growth.***Inhibiting Angiogenesis***: *GL*’s compounds may help inhibit the formation of new blood vessels that supply tumors, limiting their nutrient supply.

Studies have shown promising results regarding *GL*’s effects on various cancer types, including lung cancer, prostate cancer, breast cancer, and others [90,92]. However, it is essential to note that while some findings are encouraging, more rigorous research, including clinical trials in humans, is needed to establish its effectiveness and safety as a stand-alone cancer treatment.

*GL* is not a replacement for conventional cancer therapies; it shows promise as a potential complementary addition to cancer treatment due to its bioactive compounds and immunomodulatory effects. Ongoing research will further elucidate its role in cancer treatment, and improve our understanding of its full potential in combating this devastating disease.

### 5.1. Triple-Negative Breast Cancer

Triple-negative breast cancer (TNBC) is a particular form of breast cancer distinguished by the lack of three significant receptors: estrogen receptor (ER), progesterone receptor (PR), and human epidermal growth factor receptor 2 (HER2) [32]. These receptors are crucial for guiding targeted therapies in breast cancer treatment. However, TNBC lacks these receptors, making it more challenging to treat compared to other breast cancer subtypes [93]. TNBC constitutes approximately 15% to 20% of all diagnosed breast cancer cases, and is more commonly diagnosed in younger women, African-American women, and those with a breast cancer 1 (BRCA1) gene mutation [32,93,94]. It is known to have aggressive behavior, faster growth, and a higher likelihood of metastasis compared to other breast cancer subtypes [30]. Due to the absence of ER, PR, and HER2 receptors, TNBC does not respond to hormone therapy (such as tamoxifen or aromatase inhibitors) or targeted therapies like trastuzumab (Herceptin) [95]. As a result, chemotherapy remains the mainstay of treatment for TNBC. Various chemotherapy regimens are used to target and kill rapidly dividing cancer cells. TNBC research is ongoing to identify new treatment strategies and targeted therapies. Immunotherapy, poly (ADP-ribose) polymerase inhibitors (PARPi), and other emerging treatments are being investigated to improve outcomes for TNBC patients [32,93,94,95]. Clinical trials play a critical role in testing novel therapies for this aggressive subtype of breast cancer. TNBC patients often experience lower 5-year survival rates compared to other breast cancer subtypes due to the aggressive nature and resistance of the malignancy [93]. The aggressive behavior of TNBC leads to rapid tumor growth and a higher likelihood of metastasis. This underscores the urgent requirement for additional research to formulate new targeted therapies for TNBC. In the search for new treatment options, natural products have become an area of interest. These products possess diverse chemical structures and exhibit high specificity in their biochemical actions. As such, they form a valuable compound library for evaluating and discovering potential new drugs for TNBC and other malignancies. Studying natural products may lead to the identification of novel compounds with therapeutic potential that could complement existing treatment approaches for TNBC. *GL* has attracted interest in the field of cancer research, including for its potential effects on TNBC. Nevertheless, it is crucial to acknowledge that research in this domain is still at an early stage, and further studies are essential to ascertain the efficacy and safety of *GL*, particularly in the treatment of TNBC. Here is an overview of the current understanding:***Anticancer effects***: In vitro studies focused on MDA-MB-231 breast cancer cells highlighted the potential of *GL* extracts for inhibiting the adhesion and further migration of cancer cells through the interruption of phosphatidylinositol 3-kinase (Pl3K). Another study observed that inhibition of interleukin 8 (IL-8) secretion by *GL* extracts was associated with oxidative-stress suppression. In addition, ganoderic acids have also shown promising results in avoiding the progression of these invasive cells. This was due to the inhibition of activator protein-1 (APA-1) and nuclear factor kB (NF-kB) [96,97,98]. Regarding in vivo studies, one underscored the effectiveness of *GL* extracts in impairing tumor growth. The animal model (mice) was injected with CD44+/CD24− breast cancer stem-like cells and the results confirmed a significant reduction in tumor weight. Another in vivo study in a mice model focused on targeting the signal transducer and activator of transcription 3 (STAT3) signaling, which plays a major role in cancer stem cells maintenance [32,99].***Immunomodulation***: TNBC is distinguished by its aggressive nature and the absence of specific targeted treatment options. Immunomodulatory properties of *GL* may be relevant in the context of TNBC, as they can potentially enhance the body’s immune response against cancer cells [96,97,98]. Some studies suggest that *GL* can modulate immune cells, such as natural killer cells (NK cells) and T-lymphocytes, and enhance their activity against cancer cells.***Chemopreventive potential***: In the case of TNBC, which lacks targeted therapies, chemopreventive strategies may be particularly valuable. Some studies have suggested that GL extracts or their bioactive compounds may help inhibit the initiation or progression of breast cancer, potentially reducing the risk of developing TNBC [96,97,98].

It is crucial to emphasize that the research on GL and its potential application in treating TNBC is still limited, with most available evidence originating from preclinical studies. Robustly designed clinical trials are essential to thoroughly evaluate the safety and efficacy of *GL*, particularly in the context of TNBC, in human subjects.

### 5.2. Colon Rectal Cancer

Colon and rectal cancer, often referred to as colorectal cancer, is a type of cancer that starts in the colon or rectum and typically begins as a growth of abnormal cells in the inner lining of the colon or rectum, known as polyps [100]. These cancers are closely related due to their anatomical proximity and similar characteristics [92,93,94,95,96,97,98,99,100]. Over time, some of these polyps can develop into cancer if not detected and removed early. The precise causes of colorectal cancer are not entirely comprehended, but risk factors encompass age, family history, certain genetic conditions, diet, and lifestyle choices. Indicators of colorectal cancer might involve variations in bowel habits, persistent abdominal discomfort, blood in the stool, unexplained weight loss, and fatigue. Early detection is critical, as it allows for a higher chance of successful treatment and improved outcomes. Screening tests, such as colonoscopy, fecal occult blood test (FOBT), and sigmoidoscopy, can help detect precancerous polyps or early-stage cancers [101]. If colorectal cancer is diagnosed, the treatment options are contingent on the cancer’s stage and may encompass surgery, radiation therapy, chemotherapy, targeted therapy, and immunotherapy. *GL* has been investigated for its potential anticancer properties in colorectal cancer. However, it is important to note that research on this subject is still limited. Here is an overview of the current understanding:***Anticancer effects***: Some in vitro studies observed the anticancer potential of *GL* extracts by using human colorectal cancer cells. One was performed on an SW 480 human colorectal cancer cell line and proceeded to analyze the effectiveness of *GL* extract 1 (GLE-1), with high content in polysaccharides, and *GL* extract 2 (GLE-2), with treterpenoids, on the inhibition of cell proliferation. While both fractions showed great results in this regard, it was observed that GLE-2 had a significantly higher inhibitory activity. In addition, an in vitro study in LoVo human colon cancer cells described the promising activity of GLPs in avoiding cell migration, apoptosis induction, and activation of caspases-3, -8, and -9. Regarding in vivo studies, one performed in a mice model investigated the effect of GLP on AOM/DSS-induced colorectal cancer, and the results were very satisfactory, with decreased tumor size and cancer cells, and an extra-functional gut barrier [82,102,103].***Immunomodulation***: *GL* is known for its immunomodulatory effects, meaning it can modulate the immune system. Enhancing the immune response may be relevant in the context of colon rectal cancer, as the immune system plays a crucial role in identifying and eliminating cancer cells. Some studies suggest that *GL* can modulate immune cells, and enhance their activity against cancer cells, potentially supporting the body’s immune response to colon rectal cancer [82,102,103].***Chemopreventive potential***: In the case of colon rectal cancer, chemopreventive strategies may be valuable, especially in individuals at high risk or with a history of precancerous polyps. Some studies have suggested that *GL* extracts or its bioactive compounds may help to inhibit the initiation or progression of colon rectal cancer, potentially reducing the risk of developing this disease [82,102,103]. 

It is important to note that most of the information comes from studies conducted on cancer cell lines and animal models, making it necessary to design more human clinical studies in order to assess the safety and efficacy of *GL* in humans.

### 5.3. Other Types of Cancer

#### 5.3.1. Lung Cancer

Lung cancer originates in the lungs and has the potential to metastasize to other body parts, standing as the primary cause of cancer-related fatalities globally among both men and women [77]. The two primary categories of lung cancer are non-small cell lung cancer (NSCLC) and small cell lung cancer (SCLC) [78]. NSCLC is the predominant type, representing approximately 85% of all lung cancer instances, with SCLC constituting about 15% of cases [76].

The approach to treating lung cancer is contingent upon factors such as the type of cancer, its stage, and the overall health status of the patient. It may include surgery, radiation therapy, chemotherapy, targeted therapy, immunotherapy, or a combination of these interventions [74]. Surgical removal of the tumor may be employed for early-stage lung cancer, while advanced-stage lung cancer may require a combination of treatments.

*GL* has been the subject of scientific research for its potential effects on various health conditions, including cancer. In the context of lung cancer, some studies have explored the potential benefits of GL as a complementary or alternative approach to conventional treatments [77]. However, it is important to note that further studies are needed before any definitive conclusions can be made about its efficacy as a stand-alone treatment for lung cancer.

***Anticancer Properties***: According to the literature, to date, many in vivo and in vitro studies have been developed in order to understand the impact of GL extracts in cancer treatment. Regarding in vivo studies, mainly performed in animal models (mice), the results highlight the activity of *GL* extracts in the suppression of tumor growth, angiogenesis, and interfering with cell adhesion. Similar results were observed in in vitro studies, predominantly in human lung cancer cell lines, such as A549. One underscored the potential of ganoderic acids to trigger mitochondria apoptosis in cancer cells. In addition, GLPs have shown great inhibitory activity regarding the proliferation of vascular endothelial cells and stimulation of vascular endothelial growth factor (VEGF) production in lung cancer cells [77,78,79].***Immune System Modulation***: *GL* has the potential to boost the activity of immune cells, such as natural killer cells and T-cells. Boosting the immune response may help the body’s natural defense mechanisms in recognizing and eliminating cancer cells [72,79].***Anti-Inflammatory Effects***: Chronic inflammation has been linked to the development and progression of lung cancer. *GL*’s anti-inflammatory properties may help reduce inflammation in the lungs, potentially impacting cancer growth and progression [80].***Supporting Quality of Life***: Some studies suggest that *GL* improves the quality of life in cancer patients by reducing lung cancer-related symptoms and side effects of treatments [72,77,78,79,80].

While the research on *GL*’s effects on lung cancer is promising, it is essential to approach these findings with caution, as well as conduct further human studies.

#### 5.3.2. Prostate Cancer

Prostate cancer arises in the prostate gland, a small gland about the size of a walnut located beneath the bladder in men. The prostate gland plays a role in producing semen, the fluid that nourishes and transports sperm [80]. Preventing prostate cancer entirely lacks a definitive method, yet certain lifestyle choices may potentially mitigate the risk. These include adhering to a nutritious diet, engaging in regular physical activity, and abstaining from tobacco and excessive alcohol consumption [104]. The critical components for improving outcomes in prostate cancer involve routine check-ups and screenings to facilitate early detection. In the context of prostate cancer, there is some research exploring the use of *GL* as a complementary or alternative approach to traditional treatments [95]. Some potential ways *GL* may impact prostate cancer include the following:***Anticancer Properties***: *GL*’s bioactive compounds may exhibit antitumor effects, including inhibiting cancer cell growth and promoting apoptosis in prostate cancer cells [105]. Some in vitro studies highlight the effectiveness of GL extracts on dihydrotestoterone (DHT) inhibition and impairing cell proliferation. In addition, an in vivo study performed in an animal model (mice) verified increased mitigation of tumor cells [105,106].***Immune System Modulation***: *GL* can enhance the activity of immune cells, which play a crucial role in recognizing and eliminating cancer cells. Boosting the immune response may help the body’s natural defense mechanisms to target prostate cancer cells [107,108].***Anti-Inflammatory Effects***: Chronic inflammation has been associated with the development and progression of prostate cancer. *GL*’s anti-inflammatory properties may help reduce inflammation in the prostate, potentially impacting cancer growth and progression [105,107].***Reducing Side Effects***: Some studies suggest that *GL* may help reduce the side effects of conventional cancer treatments, such as radiation therapy and chemotherapy, in prostate cancer patients [105,107].

## 6. Application of *Ganoderma lucidum* in Nanotechnology

Nanoparticles have been explored as a promising drug delivery method due to their ability to circulate freely in the blood and escape endocytosis by cells, making them suitable for targeted drug delivery [109].

By leveraging the antitumor properties of GLPs, these nanoparticles can break biological delivery barriers, reaching tumor cells, and exerting synergistic antitumor effects [110].

While *GL* has been extensively studied in traditional medicine and pharmacology, its specific applications in nanotechnology are relatively limited and recent. In that way, the development of nanoparticle-based drug delivery systems using GLPs holds promise in enhancing the therapeutic efficacy of *GL* and its antitumor effects, offering potential advancements in cancer treatment.

### 6.1. Nanoparticle Synthesis

Researchers have investigated the application of *GL* extracts in the synthesis of nanoparticles. These extracts demonstrate the capacity to function as both reducing and stabilizing agents, enabling the production of metallic nanoparticles, including gold or silver nanoparticles [111,112]. Recent studies have also explored the use of nanoparticles to enhance the extraction, delivery, and therapeutic effects of bioactive compounds from *GL*, especially in cancer treatment. One study focused on the green synthesis of zinc oxide nanoparticles (ZnO) using *GL* extract. These nanoparticles were integrated into chitosan and polyvinyl alcohol (PVA) composites to create antimicrobial and drug delivery systems. The ZnO nanoparticles exhibited enhanced antibacterial properties, improved stability and controlled release of bioactive compounds, and potential for both cancer treatment and wound healing. Green technologies in nanomaterial synthesis provide an eco-friendly, cost-effective, and scalable approach, enhancing biocompatibility and performance for applications like drug delivery and cancer treatment [113]. Among the innovative systems developed, tocosomes stand out as a promising approach. Derived from tocopheryl phosphates, these nanoparticles have shown significant potential for delivering bioactive compounds, including hydrophilic and hydrophobic molecules. They enhance bioavailability, stabilize sensitive components, and allow for controlled release, making them suitable for nutraceutical and biomedical applications. While promising, further research is needed to optimize their use for large-scale pharmaceutical purposes [114].

The resulting nanoparticles may possess unique properties, and find applications in areas like catalysis (where a substance—catalyst—accelerates a chemical reaction without being consumed or permanently altered in the process), sensing (the process of detecting or perceiving changes in the environment or within a system through the use of sensors or sensory organs), and drug delivery [111,112].

**Nanocarriers for drug delivery**: *GL* extracts or their components have been incorporated into nanocarriers for drug delivery purposes. By encapsulating therapeutic agents within nanoscale systems, such as liposomes or nanoparticles, it is possible to enhance drug stability, improve bioavailability, and target specific tissues or cells [115].**Antimicrobial nanomaterials**: *GL* extracts have shown antimicrobial activity against various microorganisms. Researchers have explored incorporating these extracts into nanomaterials, such as coatings or films, to create antimicrobial surfaces. Such surfaces could find applications in medical devices, food packaging, and other areas where preventing microbial growth is crucial [116].**Biosensors**: *GL* extracts have demonstrated potential for use in biosensing applications. By immobilizing the mushroom extract or its bioactive compounds into nanomaterials, it is possible to create biosensors capable of detecting specific targets, such as biomarkers or pollutants, with high sensitivity and selectivity [49]. It is worth noting that the research and development of *GL* in nanotechnology are still in their early stages, and further studies are needed to explore the full potential of this mushroom in various nanotechnological applications. The synthesis of *GL* nanoparticles involves the utilization of extracts or components derived from the mushroom to produce nanoparticles with unique properties. The general steps involved in the synthesis process of nanoparticles from *GL* are the preparation of *GL* extract, reduction and stabilization of nanoparticles, and characterization and functionalization [42,111,112,116,117,118]. Techniques such as transmission electron microscopy (TEM), scanning electron microscopy (SEM), X-ray diffraction (XRD), dynamic light scattering (DLS), and Infrared Spectroscopy (FTIR) are commonly used for nanoparticle characterization [110].

Specific details of *GL* nanoparticle synthesis may vary depending on the research or study. Different extraction methods, metal ions, and reaction conditions can influence the size, shape, and properties of the resulting nanoparticles. Researchers continue to explore and optimize these synthesis methods to harness the unique properties of *GL* in nanoparticle applications.

### 6.2. Silver Nanoparticles

*GL* is often diluted in saline for intravenous administration, but it can also be administered orally [54]. However, oral administration presents certain drawbacks, such as poor stability and low bioavailability. To address these issues, novel drug delivery systems like microcapsules and microspheres have been formulated to augment the bioavailability and reduce the toxicity of GLPs [119]. The antitumor attributes of GLPs are thought to be associated with their modulation of diverse biological processes. These include the regulation of immune and inflammatory responses, induction of toxicity in tumor cells, and promotion of apoptosis. Additionally, metal nanomaterials like Au, Ag, and Pt have garnered attention owing to their distinctive chemical and optical properties, offering additional possibilities for advanced drug delivery systems [111,112]. *GL* silver nanoparticles (*GL* AgNPs) refer to silver nanoparticles (AgNPs) that are synthesized using extracts or components derived from *GL* [86]. Silver nanoparticles have stood out for their remarkable properties, including antimicrobial activity, catalytic properties, and potential applications in various fields.

The synthesis of *GL* AgNPs typically involves the following steps:**GL extract preparation**: Similar to the general process described earlier, an extract is obtained from *GL*. The extraction can be performed using solvents such as water or ethanol. The extract contains bioactive compounds that will serve as both reducing and stabilizing agents in the process of nanoparticle synthesis [116,117].**AgNPs synthesis**: The *GL* extract is mixed with a silver precursor, such as silver nitrate (AgNO_3_). The reducing agents present in the extract will promote the reduction of silver ions into silver nanoparticles. The nanoparticles form and stabilize in the final solution [42,116,117,118].**Characterization**: The synthesized *GL* AgNPs are then characterized to determine their size, shape, distribution, morphology, and other properties. Techniques such as TEM, SEM, XRD, UV–Vis and FTIR spectroscopy, and DLS can be employed to analyze the nanoparticles and assess their characteristics [42,110,117,118].**Functionalization**: If desired, the *GL* AgNPs can be further functionalized by modifying their surface. This involves the attachment of specific molecules called ligands or coatings to enhance their stability and biocompatibility, or targeting cellular capabilities for particular applications [54,120]. The specific synthesis methods and conditions may vary among different studies and researchers. The concentration of *GL* extract, silver precursor, reaction time, and temperature can all influence the size and properties of the resulting silver nanoparticles (Figure 4). It is worth noting that further research and optimization are ongoing to explore the potential applications and benefits of *GL* AgNPs in various fields, including biomedicine, catalysis, and environmental remediation.

### 6.3. Polymeric Micelles

*GL* polymeric micelles (PMs) refer to micellar structures formed by the self-assembly of polymers derived from *GL* or incorporating extracts/components from the mushroom [105]. PMs are nanoscale assemblies composed of amphiphilic block copolymers, where one block is hydrophilic and the other is hydrophobic [121]. These micelles have gained recognition for their potential uses in drug delivery, given their ability to entrap hydrophobic drugs, thereby improving their solubility and stability.

Here are the general steps involved in the synthesis of *GL* PMs:**Polymer selection**: Suitable polymers derived from *GL* or incorporating extracts/components from the mushroom are chosen. These polymers should possess amphiphilic properties, with one segment being hydrophilic and the other hydrophobic [49,122].**Polymer synthesis**: The selected polymers are synthesized using appropriate techniques, such as polymerization or modification reactions [122]. The hydrophilic and hydrophobic blocks are incorporated into the polymer structure, resulting in an amphiphilic copolymer.**Micelle formation**: The synthesized amphiphilic copolymer is then dissolved in a suitable solvent, typically aqueous solution. Due to the amphiphilic nature of the polymer, it self-assembles into micellar structures in the solution. The hydrophilic segments of the polymer form the outer shell of the micelle, while the hydrophobic segments aggregate, forming the core, and encapsulating hydrophobic drugs or other cargo [122,123,124] (Figure 5).**Characterization and functionalization**: The resulting *GL* PMs are characterized to assess their size, morphology, stability, critical micelle concentration (CMC), drug-loading capacity, and efficiency of encapsulation. Techniques such as DLS, TEM, and drug release studies are commonly employed [54,120,123]. The micelles can also be further functionalized by modifying the surface with targeting ligands or other functional moieties to enhance their specificity and therapeutic efficacy. *GL* PMs hold promise for targeted drug delivery systems, as the bioactive compounds from the mushroom may contribute to additional therapeutic effects. However, the research and development of *GL* PMs are still ongoing, and further studies are needed to explore their full potential and optimize their performance in drug delivery applications.

### 6.4. Lipid Nanoparticles

LNPs refer to nanoparticles that are formed by encapsulating *GL* extracts or components within a lipid-based delivery system [125]. They are colloidal carriers composed of lipids, and can be used for various purposes, such as drug delivery, gene delivery, and cosmetic formulations [122]. LNPs are described as particularly effective in drug delivery for cancer treatments [126]. The synthesis of *GL* LNPs generally involves the following steps:**Selection of lipids**: Lipids are chosen based on their biocompatibility, stability, and ability to form nanoparticles. Common lipids used in lipid nanoparticle formulations include phospholipids, such as phosphatidylcholine or phosphatidylglycerol, and other lipid-based materials like solid lipids or oils [119,127] (Figure 6).**Preparation of lipid solution**: The selected lipids are dissolved in an appropriate organic solvent, such as chloroform or ethanol, to form a lipid solution [127,128]. The *GL* extracts or components are incorporated into the lipid solution during this step.**Emulsification**: The lipid solution containing *GL* extracts is then emulsified with an aqueous phase, typically a buffer or water. This can be achieved through techniques like ultrasonication, high-pressure homogenization, or microfluidics, resulting in the formation of small droplets [128].**Nanoparticle formation**: After emulsification, the organic solvent is removed by evaporation or other methods, leading to the formation of LNPs encapsulating *GL* extracts [128]. The removal of the organic solvent allows the lipids to solidify and stabilize, forming nanoparticles with the *GL* components entrapped within.**Characterization**: The *GL* LNPs are characterized to determine their size, morphology, encapsulation efficiency, drug-loading capacity, and stability. Techniques such as DLS, TEM, SEM, UV–Vis and FTIR spectroscopic methods, and DSC may be employed to assess these properties [49,128].**Functionalization**: Depending on the desired application, the surface of the *GL* LNPs can be further functionalized with targeting ligands, polymers, or other surface modifications to improve their specificity, stability, or targeting properties. *GL* LNPs have the potential to enhance the delivery and bioavailability of *GL* bioactive compounds. However, specific formulation strategies and optimization processes may vary depending on the desired application and intended use of the LNPs. More research and development are needed to exploit the full potential of *GL* LNPs in various fields, including pharmaceuticals, nutraceuticals, and cosmetics.

### 6.5. Polymeric Nanoparticles

*GL* PNPs are nanoparticles formed using polymers derived from *GL* or incorporating extracts/components from the mushroom. These nanoparticles are created by self-assembling the polymers into nanoscale structures. They can be used for a variety of purposes, ranging from drug delivery to imaging and tissue engineering [129]. The synthesis of *GL* PNPs generally involves the following steps:**Polymer selection**: Polymers derived from *GL* or incorporating extracts/components from the mushroom are chosen based on their biocompatibility, stability, and ability to self-assemble into nanoparticles [129]. These polymers can include *GL*-derived polysaccharides, proteins, or modified polymers with incorporated mushroom extracts.**Polymer synthesis or modification**: The selected polymers are synthesized or modified to incorporate the desired properties for nanoparticle formation. This can involve polymerization techniques or chemical modifications to introduce hydrophilic and hydrophobic segments within the polymer structure, which are essential for self-assembly into nanoparticles [130].**Nanoparticle formation**: The synthesized or modified *GL* polymers are dissolved in an appropriate solvent to form a polymer solution. Self-assembly of the polymers occurs spontaneously due to the establishment of hydrophilic and hydrophobic interactions [123]. This results in the formation of PNPs encapsulating *GL* components or in the components integrating within the polymer matrix.**Characterization**: The *GL* PNPs are characterized to determine their size, morphology, stability, drug-loading capacity, and efficiency of encapsulation. Techniques such as DLS, TEM, SEM, and UV–Vis and FTIR spectroscopy can be employed to assess these properties [49].**Functionalization**: Depending on the desired application, the surface of the *GL* PNPs can be further functionalized with targeting ligands, polymers, or other surface modifications to improve their specificity, stability, or targeting properties. Surface modifications can also enable the attachment of imaging agents or other functionalities.

*GL* PNPs have the potential to be utilized as carriers for controlled drug release, improving the solubility and bioavailability of *GL* components, or as vehicles for targeted delivery of therapeutics. Optimizing the formulation strategies, enhancing stability, and assessing the therapeutic efficacy of *GL* polymeric nanoparticles in various applications are still challenges [131,132] (Figure 7 and Figure 8).

Preclinical studies that explore the application of *GL* in nanotechnology have shown promising results for various therapeutic purposes. These studies have utilized nanotechnology-based approaches to enhance the delivery, efficacy, and selectivity of bioactive compounds derived from *GL*.

**Enhanced Drug Delivery**: Nanoparticles loaded with *GL* bioactive compounds have been investigated for improved drug delivery in cancer therapy. In one study, polymeric nanoparticles loaded with GLPs exhibited enhanced cellular uptake and cytotoxicity against cancer cells, when compared with free polysaccharides [23,65]. The resulting NPs demonstrated sustained release of the bioactive compounds, resulting in prolonged anticancer effects.**Targeted Therapy**: Targeted delivery of *GL* bioactive compounds to cancer cells has been achieved using functionalized nanoparticles. In a preclinical study, folate-conjugated NPs encapsulating GLTs selectively targeted folate receptor-expressing cancer cells [23,65]. This targeted delivery approach improved the efficacy of the bioactive compounds and reduced toxicity to healthy cells.**Synergistic Effects**: Nanotechnology has been employed to combine *GL* bioactive compounds with other therapeutic agents, leading to synergistic effects. For example, in a preclinical study, co-encapsulation of GLTs and some chemotherapeutic drugs (e.g., Paclitaxel, Doxorubicin, Cisplatin, 5-Fluorouracil, Gemcitabine, Etoposide, and Vinblastine) within nanoparticles resulted in enhanced cytotoxicity against cancer cells compared to the individual treatments alone [23,65]. The combination therapy demonstrated improved antitumor activity and reduced drug resistance.**Immunomodulation**: Nanotechnology-based formulations incorporating *GL* bioactive compounds have shown potential for immunomodulatory effects. In a preclinical study, nanocarriers loaded with GLPs effectively stimulated immune responses, and enhanced the activation of immune cells, leading to improved anticancer immune responses [23,65]. The nanotechnology-mediated delivery facilitated the targeted modulation of the immune system.**Theranostics**: *GL*-based nanomaterials have been explored for theranostic applications, combining therapy, and diagnostics. In a preclinical study, multifunctional nanoparticles loaded with *GL* bioactive compounds were developed as theranostic agents for simultaneous cancer therapy and imaging technology [23,65]. The NPs exhibited selective tumor accumulation, efficient tumor regression, and imaging capabilities for real-time monitoring of treatment response.

These preclinical studies highlight the potential of *GL* in combination with nanotechnology for enhanced therapeutic outcomes in cancer therapy [23,65]. While they show promising results, further research is necessary to evaluate the safety, long-term effects, and clinical translation of these nanotechnology-based approaches. Nevertheless, these preclinical studies lay the foundation for future investigations, and the development of novel therapeutic strategies utilizing *GL* in nanotechnology for treatment of chronic diseases like cancer. Below, the variation in therapeutic efficacy of cancer treatment with and without nanoparticles is highlighted (Table 4).

## 7. Regulatory Issues and Clinical Trials

Transitioning from the laboratory to practical applications often encounters a complex web of regulatory challenges and rigorous clinical trials. In the case of *GL* in nanotechnology, this transition is no exception. The utilization of *GL* in nanotechnology applications within clinical trials presents several regulatory considerations that must be carefully navigated. These considerations are vital for ensuring patient safety, the efficacy of treatments, and adherence to regulatory standards. We may consider the following:**Safety and Toxicity Assessment**: Regulatory bodies require a thorough evaluation of the safety profile of *GL*-based nanotechnological products. This includes assessing potential adverse effects, toxicity, and interactions with other treatments or medications.**Standardization and Quality Control**: Ensuring the consistency and quality of *GL*-derived nanoparticles or formulations is crucial. Regulatory agencies often require standardized processes and rigorous quality control measures to maintain product integrity.**Clinical Trial Authorization**: Clinical trials involving *GL* in nanotechnology applications typically require authorization from regulatory bodies such as the United States or the EMA in Europe. Obtaining these approvals involves providing detailed documentation on the product, its manufacturing process, and preclinical data.**Data Integrity and Reporting**: Regulatory agencies expect accurate and complete reporting of clinical trial data. This includes transparency in reporting both positive and negative results, adverse events, and patient outcomes.**Good Clinical Practice (GCP)**: Adherence to GCP guidelines is essential. GCP ensures that clinical trials are conducted ethically, with patient safety in mind, and that the data collected are reliable and credible.**Post-Market Surveillance**: After clinical trials, regulatory agencies may require post-market surveillance to continue monitoring the safety and efficacy of *GL*-based nanotechnological products once they are in use by the general population.

### 7.1. Preclinical Studies

Preclinical studies are essential for enhancing our understanding of human diseases, exploring biochemical events, physiological processes, and testing new pharmacotherapies. In vitro studies are commonly used due to their ease, low cost, and minimal technical requirements. However, they may not accurately replicate the complexities of natural environments, limiting the identification of potential therapeutic compounds. In contrast, in vivo studies with animal models offer a better understanding of physiological interactions and the effects of interventions. Despite their advantages, the number of new leads from in vivo studies that progress to clinical trials remains limited. Both approaches are complementary in preclinical research for identifying potential therapies. The literature on *GL* suffers from similar limitations, including unreliable extracts, geographic variation, diverse statistical models, and complexities in dosage and mechanisms. Addressing these challenges through standardized protocols can improve the reliability and comparability of findings. In addition to efficacy, it is crucial to monitor the toxicity, adverse effects, and chronic use of test drugs or products [69,137,138].

A total of 210 articles were reviewed for preclinical studies, investigating the potential activities of *GL* in various areas [2,4,137]. Among these studies, approximately 33% utilized *GL* extract in different forms, 21% focused on isolated polysaccharides, 5% examined triterpenes, and 3% explored the effects of *GL* spore powder. The remaining 38% of studies investigated other preparations of *GL*. Regarding the study models employed, approximately 40% of the studies utilized mice, 33% used rats, 17% employed various types of cell lines, while a smaller proportion of the studies involved pigs, chickens, bacterial strains, and clinical isolates. The in vitro studies employed a dose range of 1–1000 μg/mL, while in vivo studies used doses ranging from 10 to 10,000 mg/kg [23,65]. Table 5 summarizes the most relevant preclinical studies conducted on *GL* and their related therapeutic effects.

### 7.2. Clinical Studies

Limited information is available on clinical studies specifically investigating the use of *GL* in nanotechnology-based approaches for humans. However, the information founded on clinical studies involving *GL*, and its bioactive compounds in general, may include some studies that do not specifically focus on nanotechnology. To date, no clinical studies in this regard have been conducted. Clinical studies on *GL* have explored its possible therapeutic properties in several health disorders, such as cancer. Some of these studies have investigated the administration of *GL* extracts or preparations, which may or may not involve nanotechnology-based formulations. A few examples of clinical studies involving *GL* are the following:**Cancer Therapy**: Clinical trials have evaluated the efficacy and safety of *GL* in cancer patients. These studies have explored its potential as an adjuvant therapy to conventional cancer treatments, such as chemotherapy or radiation therapy [23,58,65]. While some studies have reported positive outcomes, including improved quality of life, immune system modulation, and enhanced treatment response, the overall evidence is limited, and more rigorous studies are needed.**Immunomodulation**: Clinical investigations have explored the immunomodulatory effects of *GL* in various populations, including healthy individuals and patients with chronic diseases. These studies have explored the impact of *GL* on immune parameters, such as cytokine levels, immune cell activity, and antioxidant status [23,65,69,138,139,140,141,142]. Results have indicated potential immunomodulatory effects, but further research is needed to establish clear clinical recommendations.**Liver Health**: *GL* has been studied in clinical trials focusing on liver health, particularly in patients with hepatitis B or hepatitis C. These studies have assessed its potential hepatoprotective effects, antiviral activity, and impact on liver function. While some studies have reported positive outcomes, the evidence is still limited and larger, well-controlled trials are needed to confirm these findings [23,65,141].**Cardiovascular Health**: Cardiovascular health markers, such as blood pressure, cholesterol levels, and oxidative stress, have been assessed in clinical studies. Some trials have reported potential benefits, including improved lipid profiles and antioxidant status [17,65,142]. However, more robust clinical trials are required to establish the efficacy and safety of *GL* in cardiovascular health management.

Despite these clinical studies providing insights into the potential therapeutic effects of *GL*, the specific use of nanotechnology-based approaches in these studies may be limited. More studies are needed to explore the application of *GL* in nanotechnology-based formulations in clinical settings, assessing their safety, efficacy, and potential advantages over conventional formulations.

In the 22 clinical studies conducted on *GL*, various preparations were used, including *GL* tablets, capsules, supplements, extracts, purified polysaccharides, and polysaccharide peptides. Among these studies, 32% utilized *GL* tablets/capsules/supplements, 27% used extracts, 23% used purified polysaccharides, 9% used polysaccharide peptides, and the remaining two studies utilized supplements. The sample sizes (*n*) in these studies varied from two patients (hay fever) to 170 asymptomatic children. The administered doses ranged from 150 mg/day to 6000 mg/day [23,65].

### 7.3. Critical Assessments of the Pharmacological Activities

Clinical studies are crucial for evaluating the effectiveness and safety of medicinal products like *GL*. However, they are scarce when compared to preclinical studies, which might be due to the challenges in translating preclinical findings into clinical settings. Most of the clinical studies on *GL* have small sample sizes, which can limit the interpretation of results, and increase the risk of false positive or negative outcomes. These preliminary studies provide valuable data that can be used to design larger confirmatory studies. To establish the efficacy and safety of *GL* for marketing purposes, advanced clinical studies covering different phases (Phase I to Phase V) are necessary. It is important to have a consistent dose range, and standardized preparation methods for *GL* in clinical studies. However, there is variation in the dose range used and the reporting of mechanisms in some studies. A more systematic and reproducible approach, including sequential Phase I to Phase III studies, is needed to generate reliable data [23,65]. The source of *GL* used in clinical studies also varies in terms of geographical origin, extraction methods, and final product concentration. This variability may result in variations in the concentration of bioactive compounds, and subsequently affect the therapeutic and toxic outcomes observed. Challenges such as heterogeneity, small sample sizes, inappropriate research methodologies, lack of multicenter involvement, and inadequate statistical models have hindered the progress of *GL* as a potential conventional drug for treatment [23,65].

In summary, there is a need for well-designed clinical studies with larger sample sizes, standardized dosing, reproducible data, appropriate research methodologies, and multicenter collaboration to fully explore the potential of *GL* as a conventional drug.

*GL* has attracted significant interest in cancer therapy and other chronic diseases due to its bioactive compounds and potential health benefits. As mentioned throughout this review, in recent years, nanotechnology has emerged as a promising approach to enhance the delivery and effectiveness of therapeutic agents, including *GL* bioactive compounds in cancer treatment [143]. The combination of *GL* with nanotechnology offers exciting prospects for improving cancer therapy and patient outcomes. *GL* bioactive compounds, such as GLPs and GLTs, and other secondary metabolites, have shown promising antitumor properties in preclinical studies [56,57,58,59]. Their ability to modulate immune responses, induce apoptosis, and inhibit tumor growth makes them attractive candidates for cancer therapy. Nanotechnology-based formulations can overcome the limitations in the delivery of *GL* bioactive compounds to specific target sites, including improved delivery, increased bioavailability and stability, targeted drug delivery, and synergistic effects with other therapeutic agents [58]. Nano-sized carriers, such as nanoparticles and liposomes, offer controlled and sustained drug release, enhancing therapeutic efficacy [54,109,112,115,119,120,122,124,125,127]. Functionalized nanoparticles and liposomes offer controlled and sustained drug release, and can enhance targeted delivery to cancer cells, minimizing off-target effects and maximizing therapeutic efficacy. Furthermore, the immunomodulatory effects of *GL* can be amplified through nanotechnology, leading to enhanced activation of the immune system against cancer cells [23,65]. Nanotechnology-based approaches also offer opportunities for cancer diagnostics and imaging, facilitating early detection and personalized treatment strategies. The safety, long-term effects, and clinical translation of these approaches require thorough investigation. Additionally, the scalability, standardization, and optimization of nanotechnology-based formulations incorporating *GL* need to be addressed to ensure their practical application in clinical settings. The combination of *GL* bioactive compounds with chemotherapeutic drugs within nanoparticles has demonstrated synergistic effects in preclinical studies [90,108]. This approach has the potential to enhance cytotoxicity against cancer cells, reduce drug resistance, and minimize systemic toxicity. Furthermore, nanoparticles can be engineered with specific surface modifications or ligands to enable targeted drug delivery to tumor cells, enhancing precision medicine approaches and minimizing damage to healthy tissues. However, clinical studies specifically focusing on *GL* in nanotechnology-based approaches are currently limited.

Several nanoliposome-based therapeutics have achieved Food and Drug Administration (FDA) approval and are widely used in clinical practice. Vyxeos^®^, approved in 2017, combines daunorubicin and cytarabine for treating specific acute myeloid leukemia subtypes, offering superior survival outcomes. In addition, AmBisome^®^, a liposomal amphotericin B, treats fungal infections and leishmaniasis with reduced toxicity. Exparel^®^, approved in 2011, and Doxil^®^/Caelyx^®^, a doxorubicin formulation, have also been approved for human use. The latter targets cancers like ovarian and multiple myeloma via enhanced permeability and retention. These advancements illustrate the clinical impact of nanoliposomes technologies and potential for broader applications. While there are still no FDA-approved nanotechnology products using GL bioactive compounds, significant advancements in this area underscore its potential clinical relevance [144,145,146].

Given the promising preclinical findings, future perspectives in this field involve the development of robust clinical trials specifically focused on *GL* in nanotechnology-based approaches for cancer therapy in order to elucidate the mechanisms of action involved and optimize formulations. These trials should evaluate the safety, efficacy, and long-term outcomes of these formulations. Furthermore, efforts should be made to establish standardized manufacturing processes, quality control measures, and regulatory frameworks for these nanotechnology-based formulations.

In this way, the future perspectives in this field could involve continued research on the bioactive compounds of *GL* and their interactions with nanocarriers, for optimizing nanoformulations. Additionally, large-scale preclinical studies and well-designed clinical trials are needed to validate the efficacy and safety of *GL*-based nanomedicines in humans. Personalized nanotherapies using *GL* bioactive compounds tailored to individual patient profiles could also advance cancer treatment.

In this regard, exploring *GL* in combination with immunotherapy and targeted therapies may offer novel synergistic approaches. Collaborative efforts among researchers, clinicians, and the pharmaceutical industry are crucial to translating *GL*-based nanotherapies into clinical practice. Some examples of clinical studies are presented in Table 6.

*GL* in nanotechnology for cancer therapy holds great promise as a complementary approach to conventional treatments. Harnessing the potential of *GL*, bioactive compounds in nanoparticles and other nanocarriers present an exciting opportunity to advance cancer treatment strategies and improve patient outcomes.

## 8. Conclusions

The exploration of *Ganoderma lucidum* in the realm of nanotechnology reveals a compelling convergence of natural healing wisdom and cutting-edge science. This mushroom, known for its rich bioactive compounds, is finding new life in the world of nanotechnology. One of the most exciting aspects is the enhancement of drug delivery systems. By encapsulating *GL*’s bioactive components within nanoparticles, nanofibers, and nanocomposites, researchers are increasing their bioavailability, stability, and controlled release. This has the potential to revolutionize how medicines are delivered, making treatments more effective and reducing side effects. The integration of *GL* into nanofabrication techniques has yielded novel materials with remarkable properties. Incorporating its nanoparticles into polymeric matrices, for instance, has resulted in composite materials with improved mechanical strength, antimicrobial activity, and wound-healing properties. These materials hold immense promise for applications in tissue engineering, drug delivery systems, and antimicrobial coatings. LNPs were highlighted as the most effective drug delivery method. These nanoparticles are particularly effective due to their biocompatibility, biodegradability, and the ability to exploit the enhanced permeability and retention effect for targeted drug delivery, especially in cancer therapy. Regarding safety, they are generally considered safe for use in drug delivery due to their favorable biological properties, such as reduced systemic side effects and enhanced drug accumulation in tumor tissues, which minimizes exposure to normal tissues. However, more extensive clinical trials are necessary to fully evaluate the long-term safety and effectiveness these therapies.

Furthermore, nanoparticles loaded with bioactive compounds derived from this mushroom have exhibited potent anticancer activity by targeting cancer cells, inducing apoptosis, and inhibiting tumor growth. This represents a promising avenue for developing more effective and targeted cancer treatments. Additionally, immunomodulatory effects have been observed through nanotechnology-based formulations, which promote the activation of immune cells and enhance the body’s defense mechanisms. This has significant implications for bolstering the immune system and improving overall health. In conclusion, the integration of *GL* with nanotechnology opens up a world of possibilities for innovative solutions in healthcare and beyond. While the initial strides are promising, further research and development are warranted to fully exploit the synergistic benefits offered by this remarkable mushroom and nanotechnology. Together, they have the potential to reshape the landscape of medicine and wellness, offering new hope for improved human health and well-being.

## Figures and Tables

**Figure 2 pharmaceutics-17-00422-f002:**
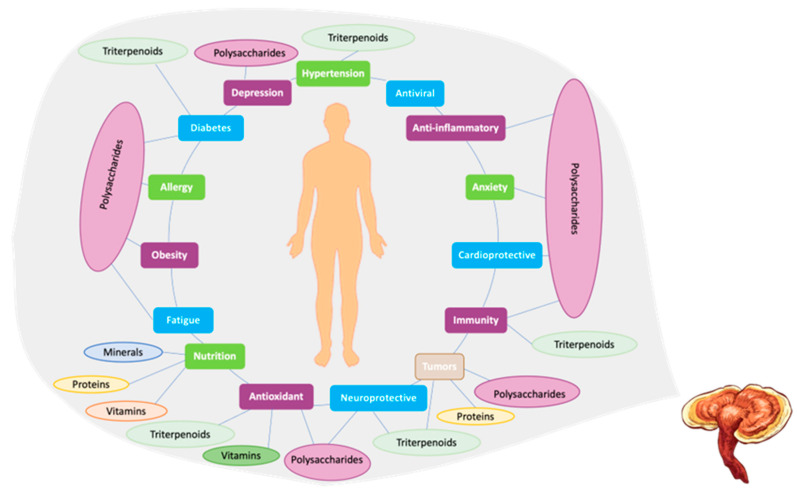
Association between the main bioactive compounds of Ganoderma lucidum and their pharmacological activities. Reproduced with permission from Ahmad, R. et al. [23], Phytotherapy Research; published by Wiley, 2021.

**Figure 3 pharmaceutics-17-00422-f003:**
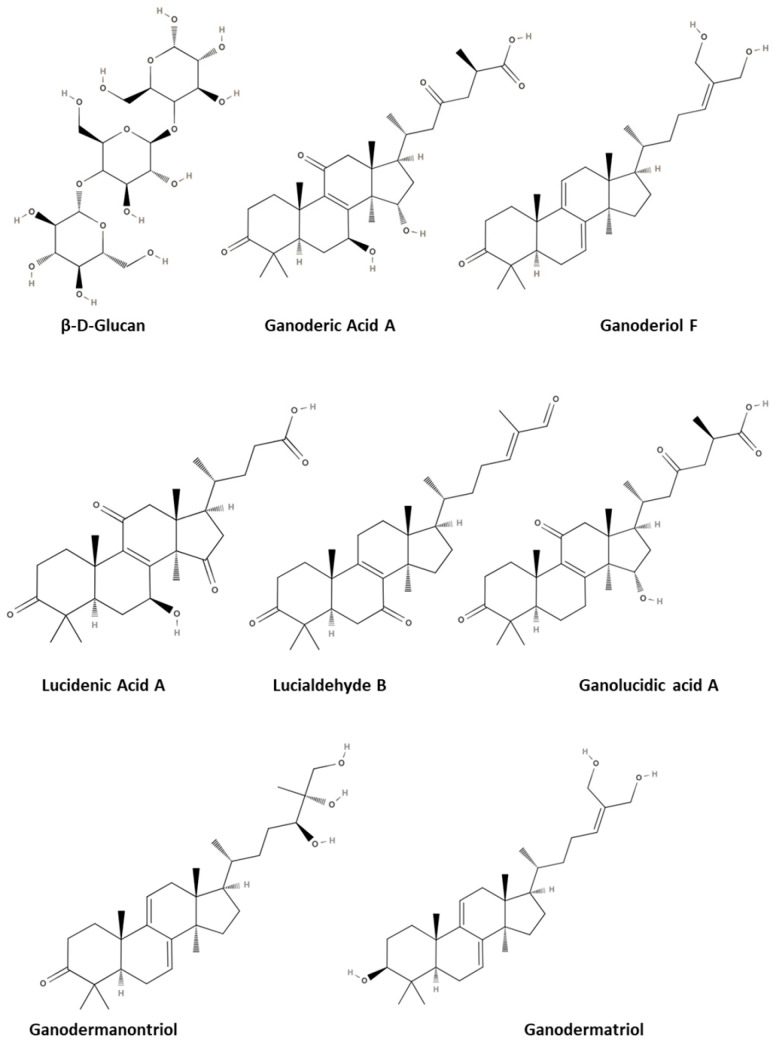
Chemical structures of main *GL* compounds.

**Figure 4 pharmaceutics-17-00422-f004:**
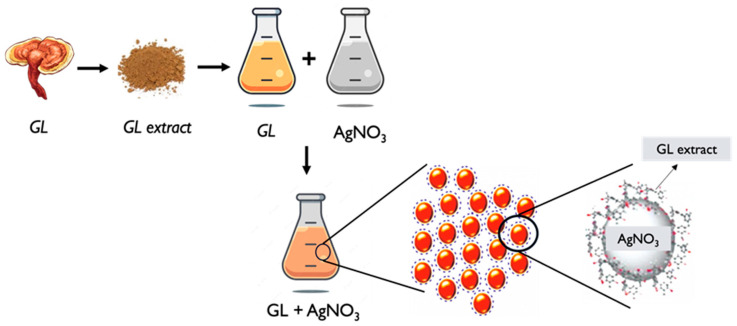
Synthesis of *GL* AgNPs.

**Figure 5 pharmaceutics-17-00422-f005:**
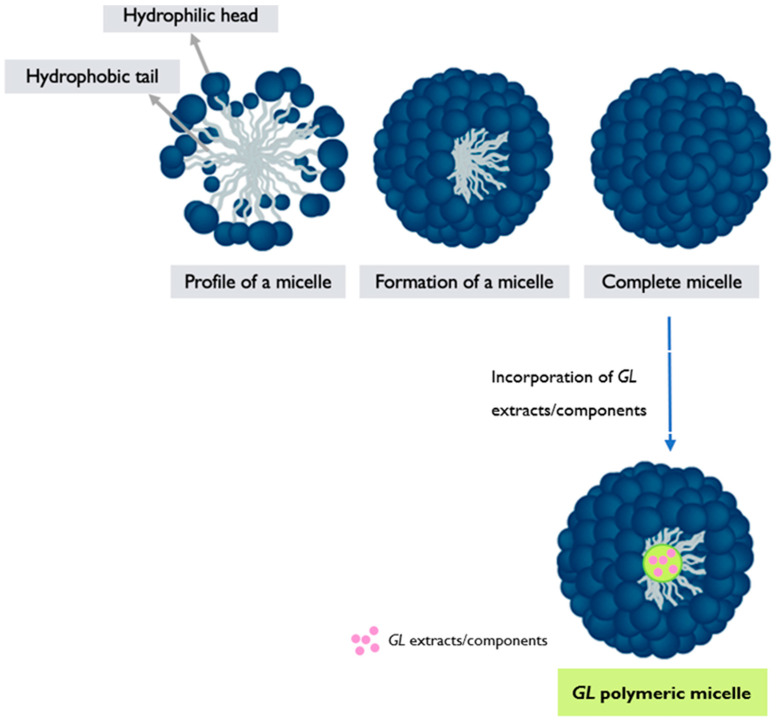
Schematic representation of the evolution from micelle formation to *GL* polymeric micelle.

**Figure 6 pharmaceutics-17-00422-f006:**
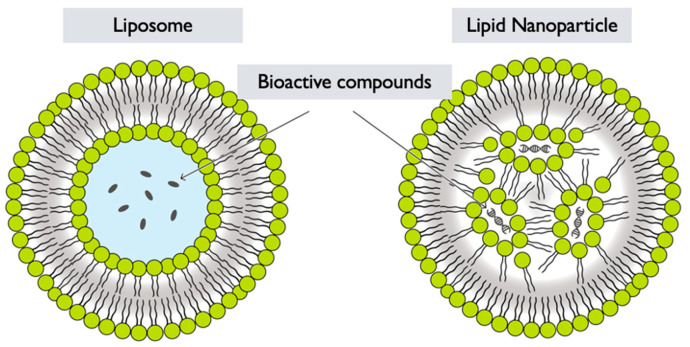
Schematic representation of liposome and lipid nanoparticle structures, highlighting their optimized design for *Gl* extracts encapsulation.

**Figure 7 pharmaceutics-17-00422-f007:**
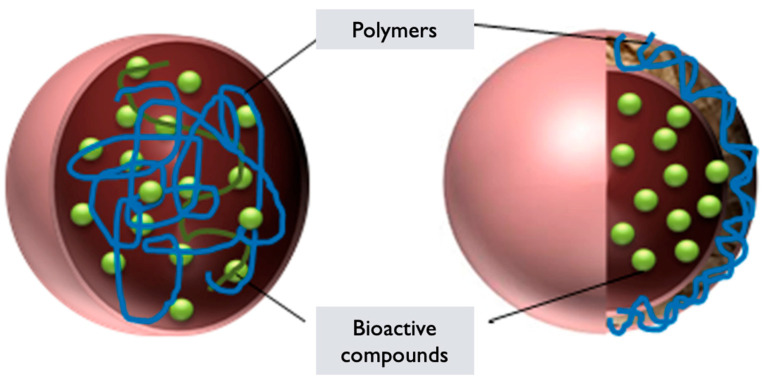
Illustration of polymeric nanoparticles encapsulating bioactive compounds, highlighting their core–shell structure.

**Figure 8 pharmaceutics-17-00422-f008:**
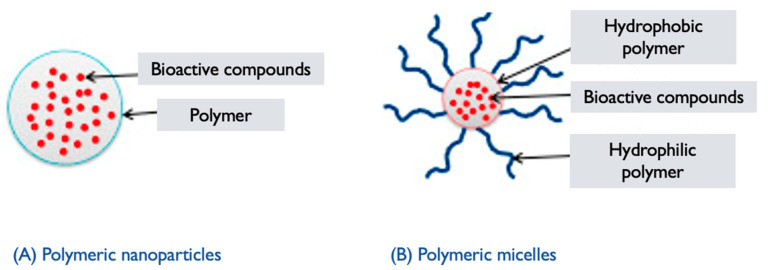
(**A**) Polymeric nanoparticle and (**B**) Polymeric micelle encapsulating bioactive compounds.

**Table 1 pharmaceutics-17-00422-t001:** Distinct steps to obtain and isolate the specific bioactive compounds from *Ganoderma lucidum* extract.

Extraction Methods		References
Hot Water extraction	The dried mushroom or mycelium is boiled in water, and the water-soluble polysaccharides are extracted. After the extraction, the solution is concentrated and then dried to obtain the polysaccharide-rich extract. Most common method for extracting polysaccharides from *GL*.	[42,43,44]
Ethanol or Methanol extraction	The dried mushroom or mycelium is soaked in ethanol or methanol to solubilize the compounds of interest. The solvent is then evaporated to obtain the extract. Most common method for extracting triterpenoids, sterols, and other secondary metabolites.	[45,46]
Supercritical Fluid Extraction	Supercritical fluid extraction uses carbon dioxide (CO_2_) as a solvent at its supercritical state (a state where it exhibits both liquid and gas-like properties). Most common method for extracting triterpenoids and essential oils.	[46,47]
Enzyme-Assisted Extraction	Enzymes can be used to enhance the extraction of specific compounds from *GL*. Most common method for extracting β-glucans from cellulases.	[45,46]
Nanoparticle- Assisted Extraction	This method involves the use of nanoparticles such as liposomes or magnetic nanoparticles to extract bioactive compounds from *GL*. The nanoparticles can selectively capture and improve the bioavailability, stability, and controlled release of the active substances, which are crucial for enhancing their therapeutic effects.	[46,48,49]
Supercritical CO_2_ with Nanotechnology	This extraction technique combines the use of supercritical carbon dioxide (SCO_2_), a highly efficient and eco-friendly solvent, with nanotechnology to improve the purity and yield of bioactive compounds. The use of nanoparticles in this process helps isolate specific compounds more effectively, while maintaining the bioactivity of the extracted molecules.	[46,47,49]
Nanoencapsulation	Nanoencapsulation involves embedding the bioactive compounds within nanocarriers, such as lipid- or polymer-based nanoparticles. This method enhances the solubility, stability, and bioavailability of the compounds, ensuring their safe and efficient delivery to target areas, such as cancer cells or the immune system.	[46]
Magnetic Nanoparticle Extraction	In this approach, magnetic nanoparticles are used to capture bioactive molecules from *GL* extracts. The nanoparticles are then separated from the mixture using a magnetic field, making the extraction process more efficient and allowing for the easy isolation of specific bioactive compounds.	[43,46]
**Separation methods**		
Solvent Extraction	A straightforward method where the dried mushroom material is soaked in a suitable solvent (such as water, ethanol, methanol, or a mixture of solvents) to extract the bioactive compounds. The solvent is then evaporated to obtain the extract.	[45,46]
Liquid–Liquid Extraction	Liquid–liquid extraction involves the partitioning of compounds between two immiscible solvents. This method can be useful for the extraction and concentration of specific compounds from the crude extract.	[50,51]
Solid-Phase Extraction (SPE)	SPE is a chromatographic technique that uses a solid-phase material (such as silica gel or other resins) to selectively adsorb and separate the target compounds from the extract.	[52]
Centrifugal Partition Chromatography (CPC)	CPC is a liquid–liquid chromatographic technique that uses a biphasic solvent system to separate compounds based on their partitioning between the two liquid phases.	[53]
High-Performance Liquid Chromatography (HPLC)	HPLC is a powerful analytical and preparative technique used to separate and purify compounds based on their chemical properties.	[45,54]
Gas Chromatography (GC)	GC is typically used for the analysis and separation of volatile compounds present in *GL*, such as essential oils.	[55]
Size-Exclusion Chromatography (SEC)	SEC is used to separate compounds based on their molecular size. It is particularly useful for the separation of polysaccharides from *GL*.	[45]
**Purification methods**		
Chromatography	Column chromatography, HPLC, and flash chromatography can be employed to separate and isolate individual compounds or groups of compounds.	[45,54]
Fractionation	The chromatographic process often generates multiple fractions containing different compounds. Each fraction can be further analyzed and tested for bioactivity to identify the most promising fractions for further purification.	[45,46]
Crystallization	For some compounds, crystallization may be employed to obtain highly purified and well-defined crystals.	[47,54]
Centrifugation	Centrifugation can be used to separate solid particles or aggregates from the purified compounds.	[45,48]

**Table 2 pharmaceutics-17-00422-t002:** Adverse effects and drug interactions reported in the literature for *GL*.

Toxicological Properties	Potential Effects	References
Allergic responses		[7,65,72]
Anticoagulants or antiplatelet medications	↑ Anticoagulant effect ↑ Prothrombin time ↑ Effects of clotting factors	[23,65,73]
Gastrointestinal bleeding or gastric ulcers	↑ Bleeding risk ↑ Gastric irritation	[23,65,72]
Hypoglycemia	↓ Blood sugar levels	[23,65,74]
Liver function	Subchronic toxicity on the liver observed in rats given *GL* extract at doses exceeding 1.2 g per kilogram of body weight.	[23,65,72,73,74,75,76,77,78,79,80]
Toxic effects on cells	↓ Cell viability at higher concentrations than those required for stimulatory results.	[23,65,74,75,77]
Antihypertensive effect	↑ Non-rapid eye movement sleep, significantly in rats, potentially linked to tumor necrosis factor-α. ↑ Effects of antihypertension drugs. ↑ Hypotension in individuals with cardiac disorder.	[39,65,73,74,77,78,81]
Toxic and teratogenic effects	In a dose- and time-dependent manner in zebrafish embryos.	[23,65,79]
Anticancer agent	↑ Toxicity when using it in conjunction with chemotherapy.	[23,57,65,72,73,74,75,76,77,78,79,80,82,83]
Antibacterial effect	↑ Activity of some antibiotics	[23,57,65,72,73,74,75,76,77,78,79,80,82,83]

**Table 3 pharmaceutics-17-00422-t003:** *Gonaderma lucidum* dosage forms and posology available [21,83].

Dosage Forms	Posology
Tablets or Capsules 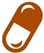	1–3 capsules/tablets of *GL* per day.
Powder 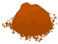	1–3 g of *GL* powder per day.
Extracts 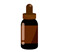	Can vary depending on the concentration and potency of the extract.
Tea or Decoction 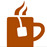	Can vary depending on the concentration, brewing time, and individual preferences.
Topical formulations 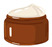	May depend on the specific formulation and intended use.

**Table 4 pharmaceutics-17-00422-t004:** Comparison of the anticancer effects of GL with and without the use of nanoparticles.

Anticancer Activity	GL	GL with NPs	References
**Induction** **of Apoptosis**	Promotes apoptosis in cancer cells through bioactive compounds like polysaccharides and triterpenes.	Targeted delivery via nanoparticles enhances apoptotic efficacy in specific tumor cells. Example: in vivo study using AuNPs focused on the incorporation of GLPs to impair breast cancer growth and induce the apoptosis of cancer cells in a tumor-bearing animal model (mice).	[32,77,79,82,102,111,112,133,134]
**Inhibition of** **Angiogenesis**	Inhibits the formation of new blood vessels that supply tumors, limiting their growth.	Sustained delivery of GL compounds improves the inhibition of angiogenesis at tumor sites. Example: in vivo study was performed in a tumor-bearing model (mice) and it was observed that *GL* polymeric NPs (loaded with triterpenoids) induced a significant decrease in blood vessels formation, thus highlighting their potential effect in this regard.	[82,90,92,102,103,135]
**Immune** **Modulation**	Stimulates immune cells like T-lymphocytes and NK cells, enhancing the anticancer immune response.	Nanoparticles enable more effective immune modulation, concentrating effects at target tissues. Example: in vitro study using mice cell lines used AuNPs for GLP delivery and it was observed that these conjugations contributed to enhancing and improving immunoregulatory properties in cancer therapy.	[90,92,96,98,107,108,111,112,135]
**Reduction in** **Inflammation**	Reduces chronic inflammation, which is linked to cancer progression.	Nanotechnology amplifies anti-inflammatory effects with controlled release at target tissues. Example: In vitro study focused on evaluating the efficacy of selenium NPs loaded with GLPs in reducing inflammation and combating cancer. The study was performed in a murine cell line and results verified great anti-inflammatory response by monitoring the secretion of pro- and anti-inflammatory cytokines.	[90,92,105,107,111,112,135]
**Tumor Specificity**	Broad and nonspecific action, affecting both healthy and cancerous cells.	Functionalized nanoparticles ensure greater specificity to cancer cells, reducing collateral damage. Example: In vitro study performed in a simulated intracellular microenvironment of cancer cells used GLP-based polymeric NPs for drug delivery. Compared to usual physiological conditions, under tumor-simulating conditions, NPs obtained greater results regarding drug release, thus highlighting their promising application in increasing bioavailability in cancer therapy.	[105,107,111,112,136]
**Bioavailability**	Limited bioavailability due to instability and poor solubility of bioactive compounds.	Nanotechnology improves stability, solubility, and efficacy of compounds in biological environments. Example: Regarding the use of polymeric GLP-NPs, an in vivo study used them to target a tumor-bearing animal model (mice) and understand related anticancer properties. In consideration of this, results found that these GLP-NPs were mainly responsible for tumor growth impairment, thus suggesting higher tumor specificity and promising therapeutic application.	[54,115,119,120,136]

**Table 5 pharmaceutics-17-00422-t005:** Preclinical studies and their therapeutic effects carried out on *GL* mushroom.

Therapeutic Effect	Action Mechanisms	Model	Reference
**Anticancer**			
In vitro In vivo	↑ CD47/CD8^+^ ratio ↑ Immune system activity ↑ Apoptosis ↑ Expression of Bax and caspase-3 ↑ mRNA expression ↑ Protein production ↑ Population of Tc-cells ↓ Activation of Akt and its downstream regulator	Cell lines related to melanoma, lung cancer, prostate cancer, colorectal cancer, breast cancer, osteosarcoma, and human prostate cancer.	[23,32,65,69,73,90,133,138]
	↓ Cellular levels; Activation of Akt and its downstream regulators; Inhibition of STAT3 signaling; cell viability, autophagy flux, Rac activity and downstream signaling pathway, osteosarcoma cell activity, and expression of anti-apoptotic proteins; ↑ Autophagy through Akt/TOR signaling, apoptosis with cell cycle arrest via NAG-1 induction, and autophagosome accumulation; ↓ Tumor volume; ↓ Growth; ↓ Metastasis; Progression and release of matrix metalloproteinases; ↑ Cytotoxicity; ↑ Apoptosis; ↑Immunomodulatory activity.	Breast cancer, mammary adenocarcinoma, ascitic tumor, cervical carcinoma, hepatoma, lung tumor, and glioma	
**Antibacterial**			
In vitro In vivo	↑ Cell permeability and leakage; ↑ Polysaccharides binding to leukocyte surfaces; Activation of Th/NK/macrophages; Upregulation of IgA/RD-5, 6/TLR4 mRNA levels; Improved attachment and permeability, increased oxidative stress, and killing of pathogens.		[23,65,139]
	↓ Firmicutes-to-Bacteroidetes ratio; ↓ Proteobacteria abundance; ↓ Levels of *Aerococcus*, *Ruminococcus*, and *Corynebacterium.*	Mice with dysbiosis and rats with type-2 diabetes	
**Anti-obesity**			
In vitro In vivo	↓ mRNA expression of SREBP-1c, C/EBPa, and PPARy; Inhibition of MAPK pathway increases energy expenditure with the inhibition of 3T3-L1 pre-adipocytes proliferation and differentiation.	Murine pre-adipocyte cells; M. miehei lipase.	[23,65,140]
	↓ Body and liver weight; ↓ Subcutaneous fat; ↑ Microbiome–gut–liver and gut–brain axes; Regulate metabolism by modulating gut microbiota composition; ↑ Levels of Clostridiales, Lachnospiraceae, Oscillospira, and Ruminococcaceae; ↓ Levels of Lactobacillus, Bifidobacterium, and Roseburia.	High-fat diet-fed; MK-fat mice.	
**Hepatoprotective**			
In vivo	↑ Antioxidant activity; ↓ Oxidative stress; Regulating key molecular pathways: FOXO4/mTOR/SIRT1; ↓ Expression of hepatic glucose regulatory enzymes, p-AMPK/AMPK, lipid peroxidation, protein oxidation, MDA, and heat shock proteins; ↓ Expression of inflammatory markers: iNOS, COX2, TNF-α, NF-KB, and IL-6; ↑ Superoxide dismutase activity, lipid peroxidation, and apoptosis; Inhibits fatty acid synthesis; ↓ Serum ALT levels indicating its potential in protecting liver health.		[23,65,141]
**Anti-dyslipidaemia**			
In vitro In vivo	↓ 3T-L1 pre-adipocytes proliferation/differentiation; ↓ Key lipid-metabolizing enzymes.		[23,65,142]
	↓ Haemorrhage/thrombosis; ↓ Stroke, cardiac necrosis; ↓ Atherosclerotic plaque; ↑ HDL-c; ↑ Total BAs.		
**Cardioprotective**			
In vitro In vivo:	↓ Cardiomyocyte necrosis; Reperfusion contracture; Antioxidant effects; Activation of PI3K/AKT signaling pathway; Modulation of specific molecular targets.		[23,65,142]
	↓ Haemorrhage/thrombosis; ↓ Stroke; ↓ Cardiac necrosis; ↓ Atherosclerotic plaque; ↑ Anti-angiogenic; ↑ Antioxidant properties.		
**Antidiabetic**			
In vitro In vivo	↓ Hepatic PECK gene expression; ↓ Glucose level; ↓ SREBP1; ↓ FAS-mRNA expression; ↓ mRNA level for gluconeogenesis enzymes and H_2_O;	Human breast adenocarcinoma cell line (MCF-7/ADR) and HepG2 cells	[23,65,142]
	↑ Glucose uptake ↑ Insulin level ↑ Hepatic glycogen level ↑ Insulin sensitivity ↑ Glycogen synthesis ↑ Glucose transport via the PI3K/Akt pathway.	Mice and rat models	
**Immunomodulatory**			
In vitro In vivo	Upregulation of immunomodulators IL-12, IF-4, IL-2, IL-6, IL-4, IL-17, TNF-a, IFN-%, granulysin, perforin, and NKG2D/NCR cell surface receptors; ↑ Production of nitric oxide (NO); Activates ERK, JNK, and p38 signaling pathways.	Mice, rats, and pigs	[23,32,65,69,73,90,100,133,138,139,140,141,142]
	Activates humoral and cellular immune responses; Promotes antigen-specific IgG production; Enhances haematopoiesis, macrophage phagocytosis, and proliferation of spleen lymphocytes and undifferentiated spleen cells; Stimulates the activity of T/B-cells, LAK cells, CD3+, CD4+, and CD8+ T-cells; Activation of NF-KB/MAPK, NK cells, NF cells, TNF activity, and cytokine secretion.		
**Anti-inflammatory**			
In vitro In vivo	↓ Expression of NF-κB, MAPK, and AP-1; ↓ Activity of G-CSF, IL-1α, MCP-5, and MIP3α; ↓ mRNA expression of CHUK and NFκB1/p150; ↓ NO, MDA, TNF-α, IL-1β, and IL-6 levels; ↓ iNOS and COX-2 expression; ↑ level of SOD.		[17,65]
	Suppression of inflammatory mediators TNF-α, IFN-γ, IL-1β, IL-6, MCP1, and hydroxyproline; ↑ Expression of keratinocyte differentiation markers; ↓ Serum Ig-E level; ↑ SOD/TOAC level.		
**Neuroprotective**			
In vivo	Downregulating caspases-3, -8, and -9; Modulation of Bcl-2/Bax ratio; Protects DNA and cell membranes from the harmful effects of radiation; ↑ Cerebral blood flow; ↓ Neuronal damage and apoptosis; Promotes mitochondrial movement; Enhances the production of anti-inflammatory cytokines; Improves spatial learning and memory-related behavior; ↓ Production of pro-inflammatory cytokines induced by Aβ and oxidative stress induced by spinal cord injury; Inhibits apoptosis caused by hydrogen peroxide, lipid peroxidation, and GSH.		[21,65]
**Anti-epileptic**			
In vivo	↓ Hippocampal neurons; ↓ Number of excitatory neurons and delays the onset of epilepsy; Prevents CA3 degeneration; ↓ Astrocytic reactivity; ↓ Levels of pro-inflammatory cytokines; ↑ Cytokines IL-1B and TNF-α; threshold for psychomotor seizures; ↑ Content of GABA; ↓ Seizures and convulsions.		[21,65]
**Sedative**			
In vivo	Inducing a hypnotic effect in rat and mice models; Promotes relaxation and sleep; Modulation of cytokines, specifically TNF-a; Sedative effects; Regulates sleep-related processes; ↓ Sleep latency; ↑ Sleep duration.		[21,65]
**Nootropic**			
In vivo	Improving cerebral blood flow, brain energy supply, memory-related neurotransmitters, and cognition; ↓ Brain cell apoptosis and ameliorates spatial memory deficits; Inhibits acetylcholinesterase activity; Antioxidant properties; Improves anterograde amnesia.		[23,65]
**Antidepressant**			
In vivo	Blocking 5-HT2A receptors; Inhibiting MAO; Antagonizing preganglionic 5-HT receptors; ↓ Depression-related activities.		[23,65]
**Anti-osteoporotic**			
In vivo	Promoting bone healing; Regeneration; ↑ Trabecular bone volume; Inhibits osteoclastogenesis and reverses bone loss; ↑ OPG/RANKL ratio; ↓ Bone differentiation; Formation of RANKL-induced osteoclast; Facilitates cross-talk between the Wnt/B-catenin and BMP/SMAD signaling pathways; Protective effects on bone.		[23,65]
**Anxiolytic**			
In vivo	↓ Anxiety levels at doses ranging between 20 and 400 mg/kg.	Swiss Albino mice	[23,65]
**Radioprotective**			
In vivo	Antioxidant and free radical scavenging properties; ↑ Levels of GSH; Protection against radiation-induced damage; ↓ Reactive oxygen species (ROS); Restoration of TNF-d production; Repair of damaged T-cells; Protection against gamma rays; Reducing DNA strand breaks and micronuclei formation; ↓ MDA levels; Promoting the recovery of SOD activity.		[23,65]

**Table 6 pharmaceutics-17-00422-t006:** Clinical studies and their therapeutic effects realized in *GL*.

Activity	Effect	References
Anticancer	↑Mitogenic reactivity to concanavalin-A and phytohemagglutinin; Lymphocyte; CD3/CD4 and natural killer cells activity; CD3/CD4/CD8/CD56, IL-2 IL-6, IFN-Y, and NK activity.	[23,65,142,147,148]
Antioxidant and hepatoprotective	↑Antioxidant activity ↓Thiobarbituric acid, 8-OH-dG. GOT and GPT levels; ↓Triglycerides; ↑HDL-c.	
Cardioprotective	↓Blood pressure and atherosclerosis; Improve chest pain/palpitation/angina pectoris; ↓Diastolic/systolic pressure, TAG, MDA, CEC, EPC levels; ↑capillary loop diameter, density, RBC velocity, and HDL-cholesterol.	
Antidiabetic	↓Cell resistance to insulin and HbA1c, FPG, and PPG values; The antiplatelet effect of *GL*, though contains a high level of adenosine; Lack of effect on platelets aggregation.	
Anti-histaminic	Most symptoms were relieved in hay fever patients due to restored normal balance between Th1 and Th2.	
Antiviral	Inhibition of virus replication in hepatitis-B and HIV patients; ↓HBeAg. HBV, DNA, and liver enzymes.	
Immunomodulatory	↑CD3+, CD4+, CD8+ T cells.	
Anti-fibromyalgia	Aerobic endurance was improved along with lower body flexibility and velocity via the antioxidant effect of *GL*.	
Anti-Alzheimer’s	↓Ab, 3, 4-methylenedioxyamphetamine, Fasl, caspase-3, and tau hyperphosphorylation.	
Anti-macular degeneration	Improvement of pre-ganglionic retinal elements in age-related macular degeneration patients with an increase in mfERG R1 and R2, and RADs.	

## Data Availability

No new data were created or analyzed in this study. Data sharing is not applicable to this article.

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
