# Peer review of "Unlocking the Potential of Ganoderma lucidum (Curtis): Botanical Overview, Therapeutic Applications, and Nanotechnological Advances"

_pharmaceutics, 2025, doi:10.3390/pharmaceutics17040422_

Round 1

Reviewer 1 Report (Previous Reviewer 2)

Comments and Suggestions for Authors

The new version of the paper is far better than previous one but still suffers  the shortcomings of previous one.

- First, is to huge and Authors could resign from some parts of text without harm to the manuscript - mostly by removal of obvious informations;

- Second, there is to many repetitions of information - for example, there is no need to repeat so many times how profitable GL is and how it is useful as source of different compounds;

- Third, important paragraph 4.1. is chaotic. Please order the stream of discussion by depicting activity of the certain group of metabolites. For eample, polysaccharides are discussed and then terpenic compounds with coming back to polysaccharides;

- Fourth, if forms of GL are discussed (paragraph 4.3.) it should be mentioned if they are available as drugs (then producer should be given), diet supplements or available out-of-cantour. If the form is only under academic study it also should be stated.

 - Fifth, paragraph 5 requires reorganization. First, general action of GL as anticancer agent should be described and then more detailed  studies considering certain  cancer types with data documented for specific cells or specific animal or clinical or animal studies should be given. For example sentence (line 640) "Some studies suggest that GL can modulate immune cells, andenhance their activity against cancer cells, potentially supporting the body's immune response to colon rectal cancer" is valid for all the cancers while for colon cancer it is only a speculation.

Sixth, if are there studies on individual compounds they should be also mentioned.

Seventh, chemical structures of the most important compounds should be given as Figure.

Eight, Table 1 should be changed - it is to big shortcut.

There are also some errors, which should be carefully corrected.

Author Response

Reviewer 1:

The new version of the paper is far better than previous one but still suffers the shortcomings of previous one.

- First, is to huge and Authors could resign from some parts of text without harm to the manuscript - mostly by removal of obvious informations;

Answer: Thank you so much for reviewing our manuscript. We made an effort to revise all manuscript and improve its organization in order to provide the reader a clear understanding of the main theme and remove obvious information. All modifications are highlighted in rose in the text.

- Second, there is to many repetitions of information - for example, there is no need to repeat so many times how profitable GL is and how it is useful as source of different compounds;

Answer: Thank you very much for your pertinent criticism. We share the reviewer’s opinion regarding the repeated information on GL and its valuable characteristics. Therefore we adjusted some parts of the text to improve the quality of our work and avoid constant information repetition.

- Third, important paragraph 4.1. is chaotic. Please order the stream of discussion by depicting activity of the certain group of metabolites. For eample, polysaccharides are discussed and then terpenic compounds with coming back to polysaccharides;

Answer: Thank you so much for your pertinent criticism. We understand the reviewer’s opinion regarding section 4.1. In this regard, the whole section was reorganized in order to provide a clear and objective reading. We also added sections 4.1.1 and 4.1.2 to focus, separately, on the major components of GL (polysaccharides and triterpenoids).

- Fourth, if forms of GL are discussed (paragraph 4.3.) it should be mentioned if they are available as drugs (then producer should be given), diet supplements or available out-of-cantour. If the form is only under academic study it also should be stated.

Answer: Thank you very much for your pertinent criticism. We share the reviewer’s opinion regarding the relevance of mentioning available GL-based drugs, diet supplements or out-of-cantour. In consideration of this, we proceeded to research and found some information on marketed formulations. The manuscript was corrected as follows:

(Section 4.3.)

“To date, these formulations are marketed, mostly, as dietary supplements. Companies such as Mountain Rose Herbs, BulkSupplements and Bio-Botanica provide GL extracts and powders, while others as Foodicine focus on tablets and capsules. Besides, topical GL-based formulations are also commercialized, for example by TRI-K Industries.”

The added references were:

  • https://www.foodicine.co.in/contract-manufacturing-ganoderma-lucidum-reishi-mushroom-extract-nutraceuticals.
  • https://mountainroseherbs.com/reishi-http://mountainroseherbs.com/reishi-mushroom?srsltid=AfmBOopSLXsAhTc5FLjLwo5gH6WX7h_u9bnR1zUtxzFfWzRFazWt6bYs.
  • https://www.bio-botanica.com/product/reishi-ganoderma-lucidum-mushroom-extract/
  • https://www.bulksupplements.com/pt/products/podeextratodecogumeloreishi?srsltid=AfmBOooFiFIOLWGlOfTOaElH8NeFkeVzd4NgwOw_mf2-AEPyWBpaX8b_
  • https://www.tri-k.com/solutions/p/naturepep-ganoderma

 - Fifth, paragraph 5 requires reorganization. First, general action of GL as anticancer agent should be described and then more detailed  studies considering certain  cancer types with data documented for specific cells or specific animal or clinical or animal studies should be given. For example sentence (line 640) "Some studies suggest that GL can modulate immune cells, andenhance their activity against cancer cells, potentially supporting the body's immune response to colon rectal cancer" is valid for all the cancers while for colon cancer it is only a speculation.

Answer: Thank you so much for your comment. We fully agree with the reviewer’s opinion and pertinent suggestions. Therefore, we adapted this section in order to provide the reader a complete understanding on documented studies. All modifications are highlighted in rose in the manuscript. We also added some references to support the information mentioned in the following sections:

(Section 5.1. Triple negative breast cancer)

  • Rios-Fuller, T. J.; Ortiz-Soto, G.;  Lacourt-Ventura, M.;  Maldonado-Martinez, G.;  Cubano, L. A.;  Schneider, R. J.; Martinez-Montemayor, M. M., Ganoderma lucidum extract (GLE) impairs breast cancer stem cells by targeting the STAT3 pathway. Oncotarget 2018, 9 (89), 35907.

(Section 5.3. Prostate cancer)

  • Zhou, X.; Wang, J.;  Guo, Y.;  Lai, H.;  Cheng, S.;  Chen, Z.;  Li, H.;  Li, Q.; Mao, X., Water extract of sporoderm-broken spores of Ganoderma lucidum elicits dual antitumor effects by inhibiting p-STAT3/PD-L1 and promoting ferroptosis in castration-resistant prostate cancer. Journal of Functional Foods 2024, 113, 106018.

Sixth, if are there studies on individual compounds they should be also mentioned.

Answer: Thank you so much for your pertinent suggestion. We agree with the reviewer’s opinion and completed the manuscript with detailed information on in vitro and in vivo studies involving GL extracts. According to the literature, and after an extensive research on this theme, most of them focused on the use of GLPs, ganoderic acids and triterpenes.

Seventh, chemical structures of the most important compounds should be given as Figure.

Answer: Thank you very much for your pertinent criticism. We agree with the reviewer’s suggestion and look forward to improve the quality of our manuscript. Therefore, we elaborated a new Figure (Figure 3 in revised version), which illustrates the most important compounds present in GL. The added references were:

- https://pubchem.ncbi.nlm.nih.gov/compound/Beta-Glucan. (accessed 07/02/2025 at 3pm).

- https://pubchem.ncbi.nlm.nih.gov/compound/Ganoderic-acid-A. (accessed 07/02/2025 at 5pm).

- https://pubchem.ncbi.nlm.nih.gov/compound/14109375. (accessed 07/02/2025 at 2pm).

- https://pubchem.ncbi.nlm.nih.gov/compound/10343868. (accessed 07/02/2025 at 5pm).

-https://pubchem.ncbi.nlm.nih.gov/compound/3001811#section=2D-Structure. (accessed 07/02/2025 at 6pm).

- https://pubchem.ncbi.nlm.nih.gov/compound/Ganodermatriol#section=2D-Structure. (accessed 05/02/2025 at 1pm).

- https://www.chemspider.com/Chemical-Structure.16737918.html. (accessed 06/02/2025 at 4pm).

- https://www.caymanchem.com/product/39775. (accessed 07/02/2025 at 5pm).

Eight, Table 1 should be changed - it is to big shortcut.

Answer: Thank you so much for your comment. We share the reviewer’s opinion and, considering  comments of reviewer 1 and 2, found interesting to complete the information of Table 1 and add it to Figure 1.

There are also some errors, which should be carefully corrected.

Answer: Thank you so much for your pertinent comment. We made an effort to improve the quality of our work by revising all manuscript, reorganize it and meet the expectations of the reviewers.

Reviewer 2 Report (New Reviewer)

Comments and Suggestions for Authors

The submitted review entitled:  A Review of the Application of Ganoderma lucidum (Curtis) P.  Karst. in traditional and modern therapies, presents a very interesting and comprehensive look at Ganoderma lucidum as an important mushroom. Yet the following pints should be revised before further steps:

Title

Remove P.  Karst., unless it is very necessary, as it is confusing. The full name can always go to: (3. Ganoderma lucidum: botanical overview).

Abstract

-The objective of this review should be mentioned clearly in the first part of the abstract

-The abstract is very long and narrative. It should be summarized to the least important points and landmarks covered in the study.

Introduction

-There is ample of information which sometimes tends to be redundant. Example line 85 to 91 which can be summarized in the beginning of the introduction.

-Add (geographical zone) if possible to table 1

-Can authors make connection/ unification between Figure 1 and table 1 if possible?

-Line 273: specify which traditional system if possible

-Table 2 needs referencing

-Rewrite the legend of Figure 2

Line 484: should be: GL can be prepared in various dosage…

-Figures and table are interesting including table 4.

-Line 726: (A total of 210 articles were reviewed) when you specify the number of articles in this part, this means it should also be specified for every part of the study. It is suggested to make a chart at the beginning of the study showing how many articles reviewed, and how many taken for each part.

-Complete the legend in figures 4, 5 and 6

-Finally authors should try to avoid any redundancies in the review study.

Author Response

Reviewer 2:

The submitted review entitled:  A Review of the Application of Ganoderma lucidum (Curtis) P.  Karst. in traditional and modern therapies, presents a very interesting and comprehensive look at Ganoderma lucidum as an important mushroom. Yet the following pints should be revised before further steps:

Title

Remove P.  Karst., unless it is very necessary, as it is confusing. The full name can always go to: (3. Ganoderma lucidum: botanical overview).

Answer: Thank you so much for revising our work. We share the reviewer’s opinion regardinf the mention to “P. Karst.” in the title. Considering that this title has also been changed according to suggestion of previous reviewers, we suggest the modification for “Unlocking the Potential of Ganoderma lucidum (Curtis): Botanical overview, Therapeutic Applications and Nanotechnological Advances”. Thus, we can meet the opinion of different reviewers.

Abstract

-The objective of this review should be mentioned clearly in the first part of the abstract

Answer: Thank you so much for your comment. We agree with the reviewer’s opinion regarding the abstract organization. Therefore, we proceeded to some modifications in order to provide the reader a better understanding of the main objective of this review. The abstract was modified as follows:

(Abstract)

“Ganoderma lucidum (GL), commonly known as the "Lingzhi" or "Reishi" mushroom, has long been recognized for its potential health benefits and medicinal properties in traditional Chinese medicine. In this regard, the main objective of this review was to explore the unique potential of GL in traditional and innovative therapies, such as cancer treatment. In recent years, the emerging field of nanotechnology has opened up new possibilities in order to use the remarkable properties of GL at the nanoscale. The unique combination of bioactive compounds present in GL, such as triterpenoids, polysaccharides, and peptides, has inspired interest in leveraging their therapeutic potential through nanotechnological approaches. Nanotechnology-based strategies have been investigated for the efficient extraction and purification of bioactive compounds from GL. Additionally, nanocarriers and nanoformulations have been developed to protect these sensitive bioactive compounds from degradation, ensuring their stability during storage and transportation. The use of GL-based nanomaterials has shown promising results in several biomedical applications, namely due to their anticancer activity by targeting cancer cells, inducing apoptosis, and inhibiting tumor growth. GL, combined with the potential of nanotechnology, presents an exciting frontier in the development of novel therapeutic and biomedical applications. Nevertheless, further research and development in this interdisciplinary field are warranted to fully exploit the synergistic benefits offered by GL, and nanotechnology, ultimately leading to innovative solutions for human health and well-being. Future prospects in this field include developing robust clinical trials focused on GL nanotechnology-based cancer therapies, to clarify mechanisms of actions and optimize formulations.”

-The abstract is very long and narrative. It should be summarized to the least important points and landmarks covered in the study.

Answer: Thank you ver much for your criticism. As we agree with the reviewer’s suggestion regarding the length of the abstract, we made an effort to exclude irrelevant information and focus on the main objective of our work.

Introduction

-There is ample of information which sometimes tends to be redundant. Example line 85 to 91 which can be summarized in the beginning of the introduction.

Answer: Thank you so much for your suggestion. We share the reviewer’s opinion and made an effort to eliminate repetitions and redundant information. All manuscript was revised and reorganized in order to avoid this in all sections of the text.

-Add (geographical zone) if possible to table 1

Answer: Thank you very much for your suggestion. We share the reviewer’s opinion regarding the importance of geographical zone. However, as we tried to make a connection between Table 1 and Figure 1, in order to meet the comments of the other reviewers, we thought it would become very extensive. To do this, Table 1 had to be deleted. Therefore, we include the information on color and taste/property related to different Ganoderma species. This modifications can be observed in Figure 1.

-Can authors make connection/ unification between Figure 1 and table 1 if possible?

Answer: Thank you so much for your pertinent suggestion. We agree with the reviewer’s opinion and, as mentioned before, made a unification between Figure 1 and Table 1 (now, Figure 1).

-Line 273: specify which traditional system if possible

Answer: Thank you very much for your comment. The traditional practices reffered to Traditional Chinese Medicine, Japanese and Korean Traditional Medicine. The manuscript was corrected as follows:

(Section 3.2. Uses in traditional medicine)

“The GL mushroom has long been recognized in traditional practices, such as Traditional Chinese Medicine, Japanese and Korean Traditional Medicine, as a "tonic for promoting longevity" and well-being, and has gained recognition as a valuable medicinal resource in the healthcare system”

-Table 2 needs referencing

Answer: Thank you so much for your comment. We agree with the reviewer’s suggestion and completed references of Table 2 (now Table 1 in the revised manuscript version). Added references were:

- Nguyen, V. P.;  Le Trung, H.;  Nguyen, T. H.;  Hoang, D.; Tran, T. H., Advancement of Microwave-Assisted Biosynthesis for Preparing Au Nanoparticles Using Ganoderma lucidum Extract and Evaluation of Their Catalytic Reduction of 4-Nitrophenol. ACS Omega 2021, 6 (47), 32198-32207.

- Karimi, M.;  Raofie, F.; Karimi, M., Production Ganoderma lucidum extract nanoparticles by expansion of supercritical fluid solution and evaluation of the antioxidant ability. Sci Rep 2022, 12 (1), 9904.

- Li, S.;  Hou, W.;  Li, Y.;  Liu, Z.;  Yun, H.;  Liu, Q.;  Niu, H.;  Liu, C.; Zhang, Y., Modeling and optimization of the protocol of complex chromatography separation of cyclooxygenase-2 inhibitors from Ganoderma lucidum spore. Phytochemical Analysis 2023, 34 (4), 431-442.

- Li, Y.-b.;  Wang, J.-l.; Zhong, J.-J., Enhanced recovery of four antitumor ganoderic acids from Ganoderma lucidum mycelia by a novel process of simultaneous extraction and hydrolysis. Process Biochemistry 2013, 48 (2), 331-339.

- Wubshet, S. G.;  Johansen, K. T.;  Nyberg, N. T.; Jaroszewski, J. W., Direct 13C NMR Detection in HPLC Hyphenation Mode: Analysis of Ganoderma lucidum Terpenoids. Journal of Natural Products 2012, 75 (5), 876-882.

- Yang, M.;  Dai, J.;  He, M.;  Duan, T.; Yao, W., Biomass-derived carbon from Ganoderma lucidum spore as a promising anode material for rapid potassium-ion storage. Journal of Colloid and Interface Science 2020, 567, 256-263.

-Rewrite the legend of Figure 2

Answer: Thank you so much for your pertinent criticism. We agree with the reviewer’s opinion regarding legend of Figure 2. All modifications are highlighted in pink and the legend was corrected as follows:

Figure 2. Association between the main bioactive compounds of Ganoderma lucidum and their pharmacological activities.”

Line 484: should be: GL can be prepared in various dosage…

Answer: Thank you very much for your comment. The manuscript was corrected as follows:

(4.3. Dosage forms and posology)

“GL can be prepared in various dosage forms and consumed on different routes.”

-Figures and table are interesting including table 4.

Answer: Thank you very much for your comment. We're delighted that you appreciate them. Thank you for your motivation, which is so important during the revision process and recognizes the effort we dedicate to our work.

-Line 726: (A total of 210 articles were reviewed) when you specify the number of articles in this part, this means it should also be specified for every part of the study. It is suggested to make a chart at the beginning of the study showing how many articles reviewed, and how many taken for each part.

Answer: Thank you very much for your pertinent comment, which we understand perfectly. First of all, we would like to explain to the reviewer that the information refered to was part of section 7 in the initial version of the manuscript. Due to suggestions from other reviewers, we have changed the location of this information, which in fact makes it out of context, because it only refers to preclinical studies and their therapeutic effects carried out on GL mushroom (and not only for prostate cancer).

Therefore, in order to respond to the reviewer's comment, we have put the information about preclinical studies back in section 7 (in particular 7.1). The aim is simply to present an overview of the articles researched on preclinical studies with GL and their therapeutic applications, in essence to present a general summary.

We understand the reviewer's comment, but in fact this information was not related to the prostate cancer section (where it was previously), but to the various therapeutic applications we found during this review process.

We thank the reviewer again for the pertinent comment, which allowed us to put the information back where it belongs, thus justifying the presentation of the percentage data of the studies in general only in this section of the manuscript (both for preclinical and clinical studies as can be observed).

-Complete the legend in figures 4, 5 and 6

Answer: Thank you so much for your suggestion. We agree with the reviewer’s opinion regarding the legend in Figures 4, 5 and 6 (now Figures 5, 6 and 7 in the revised manuscript version). The legends were completed as follows:

Figure 5. Schematic representation of micelle formation.”

Figure 6. Schematic representation of liposome and lipid nanoparticle structures, highlighting their optimized design for Gl extracts encapsulation.”

Figure 7. Illustration of polymeric nanoparticles encapsulating bioactive compounds, highlighting their core-shell structure.”

-Finally authors should try to avoid any redundancies in the review study.

Answer: Thank you very much for your pertinent comment. As we mentioned before, all manuscript was revised in order to improve its quality, and provide the reader a better and clear understanding on the main objective of our work. We have revised the article to avoid some redundancies. All the changes are highlighted in the revised version of the manuscript.

Round 2

Reviewer 1 Report (Previous Reviewer 2)

Comments and Suggestions for Authors

The novel version of thios paper is far better than previous one.Although it is still narrative review and contains a number of informations, which are quite obvious,  it is now suitable to be published. Now the reveiew is easy to read and interesting. However, I do propose to introduce some (minute ones) corrections, which would be beneficial to the paper. They re as follows:

1./ line 294: does it mean that nucleotides are half by half GMP and XMP? It should be explained;

2./ Figures 1 & 2 could be improved. There is something wrong with them: fonts are of variable thickness lines are not clear cut;

3./ Figure 5 remains simple micelle. Some details should be given at the Figure to differentiate polymeric and classic micelles;

4./ TAble 4, column "GL with NPs": The information there is this column is very general and inconcrete - should be more detailed (for example what kind of nanoparticles, what kind of sustainable model and what kind of biological model). Otherwise it not provides information at all.

Author Response

Reviewer 1

The novel version of thios paper is far better than previous one.Although it is still narrative review and contains a number of informations, which are quite obvious,  it is now suitable to be published. Now the reveiew is easy to read and interesting. However, I do propose to introduce some (minute ones) corrections, which would be beneficial to the paper. They re as follows:

1./ line 294: does it mean that nucleotides are half by half GMP and XMP? It should be explained;

Answer: Thank you so much for your revision and pertinent suggestions. We share the reviewer’s opinion regarding the lack of explanation on the presence of these nucleotides in GL. Therefore, we added some information to this section in orther to provide a better understanding on this subject. Although recent studies do not specify if GMP or XMP were found in the same proportion in GL extracts, it is known that these nucleotides are present in higher levels in matured basidiocarps. Therefore, the manuscript was corrected as follows:

Section 3.3.

“5’ - guanosine monophosphate and 5’ - xanthosine monophosphate were found in both young and mature forms of the mushroom, although in higher proportion in matured basidiocarps ”

2./ Figures 1 & 2 could be improved. There is something wrong with them: fonts are of variable thickness lines are not clear cut;

Answer: Thank you so much for your comment. We agree with the reviewer’s suggestion and made an effort to correct and improve the quality of Figures 1 and 2.

3./ Figure 5 remains simple micelle. Some details should be given at the Figure to differentiate polymeric and classic micelles;

Answer: Thank you very much for your pertinent comment. As we share the reviewer’s opinion regarding Figure 5, we made some modifications in order to provide the reader with a clear differentiation of classic and polymeric micelles. This is also highlighted in Figures 7 and 8.

4./ TAble 4, column "GL with NPs": The information there is this column is very general and inconcrete - should be more detailed (for example what kind of nanoparticles, what kind of sustainable model and what kind of biological model). Otherwise it not provides information at all.

Answer: Thank you so much for your comment. We agree with the pertinent suggestion provided by Reviewer 1 and, thus, proceeded to correct and clarify column “GL with NPs” from table 4. We added some examples to support information related to potential activities of GL-based NPs in cancer therapy. Thus, we strive to enhance the quality of our work and meet the rigor required for future publication. For a better understanding, we added the following references to this section:

- Wang, M.; Yu, F., Research Progress on the Anticancer Activities and Mechanisms of Polysaccharides From Ganoderma. Front Pharmacol 2022, 13, 891171.

- Gao, X.; Homayoonfal, M., Exploring the anti-cancer potential of Ganoderma lucidum polysaccharides (GLPs) and their versatile role in enhancing drug delivery systems: a multifaceted approach to combat cancer. Cancer Cell Int 2023, 23 (1), 324.

- Zheng, D.;  Zhao, J.;  Tao, Y.;  Liu, J.;  Wang, L.;  He, J.;  Lei, J.; Liu, K., pH and glutathione dual responsive nanoparticles based on Ganoderma lucidum polysaccharide for potential programmable release of three drugs. Chemical Engineering Journal 2020, 389, 124418.

Reviewer 2 Report (New Reviewer)

Comments and Suggestions for Authors

Thank you

Author Response

Thank you.

Answer: We thank you for taking the time to review our work and for your pertinent comments. Your suggestions have improved the quality of the article and we are grateful for that. We are very happy to have met your expectations.

This manuscript is a resubmission of an earlier submission. The following is a list of the peer review reports and author responses from that submission.

Round 1

Reviewer 1 Report

Comments and Suggestions for Authors

The manuscript 'A Review of the application of Ganoderma lucidum (Curtis) P. Karst. in Nanotechnology for the treatment of cancer' ontains a comprehensive review of the traditional use and research on the therapeutic properties of Ganoderma lucidum, the methods of obtaining the active principles of its extracts, ending with a description of its potential in the treatment of various types of cancer, as well as its combination with nanoparticles. 

More than half of the text is devoted to the general description and traditional uses, and the final part is devoted to a general description of nanoparticles as transport and distribution vectors and some examples of possible treatments for certain types of cancer.

A more appropriate title would be "A Review of the application of Ganoderma lucidum (Curtis) P. Karst. in traditional and modern therapies"

In general, the text is very repetitive and full of generalities and could be significantly reduced. Specifically, 

 1. Introduction, the text should be shortened and expressed in a more concise and direct way about the background and concrete objectives of the review. 

3. Ganoderma lucidum: botanical overview, characterization, uses in traditional medicine and chemical studies, should be significantly reduced by eliminating all explanations referring to the use of known techniques or methodologies; it is sufficient to mention them and to refer to the products obtained in each case.

In general, sections 4, 5, 6 and 7 can be reduced by eliminating trivial explanations of methodologies and avoiding repetitions of content in text and tables.

Once these modifications to the text are satisfactorily carried out and the manuscript gains in conciseness and clarity it should be reconsidered for publication. 

Comments on the Quality of English Language

In general the manuscript is readable.

One careful reading would be necessary to detect and correct some minor spelling mistakes.

Reviewer 2 Report

Comments and Suggestions for Authors

There is a lot of countries where mushrooms growing on trees are used medicinally. Thus the review on that subject is needed. Authors present a huge review, presumably on that problem but in fact on very general subjects. This review generally presents obvious information, which is not interesting to eneral reader.  In opposition to he subject announced by te title ithere is a little information. Thus, the paper in half looks like studnet book and i n part as advertisement of drugs based on Ganodermalucidum.

I

Reviewer 3 Report

Comments and Suggestions for Authors

In the current review article (pharmaceutics-3112222), Figueiras and collaborators aim to deliver a comprehensive overview of the potential utility of nanodelivery systems loaded with bioactives from Ganoderma lucidum for cancer treatment, either as a single approach or in combination with conventional chemotherapeutic drugs. The proposed aim and content of the manuscript clearly fit within the scope of Pharmaceutics. However, as detailed below, several issues preclude the acceptance of the manuscript.

A) While the authors clearly defined the main aim of the review, i.e., to cover the design of nanoformulations for optimized delivery of bioactive compounds from G. lucidum to target tumours, the conceptualization and organization of the manuscript have several limitations:

1. The authors did not attempt to select and provide a critical overview of the available data on the anticancer potential of G. lucidum-based nanosystems. Instead, they apparently selected data in a random manner, frequently including content that falls outside the scope of the review, as exemplified below:

Table 2: The extraction, fractionation, and purification procedures correspond to conventional methods applicable to all natural products, rather than being tailored to the specific extraction and purification of bioactives from G. lucidum.

Lines 70-75; 744-942: Similarly, the procedures to obtain and the description of the design of nanoformulations are broad and not specifically focused on delivering anticancer bioactives from G. lucidum, nor do they consider the particularities of a fungal-based natural matrix.

Lines 410-418; 759-763; Table 6; Table from pages 41-42: It is unclear why the authors presented such vast data on bioactive effects unrelated to properties that might improve outcomes in cancer therapy, such as cardiovascular, antidiabetic, sedative, and neuroprotective effects.

Lines 571-718: The authors cover purely pedagogical and general information on the physiopathology and clinical outcomes of different types of cancer, which falls outside the scope of a review article.

2. The selected data on the anticancer effects of bioactives or extracts of G. lucidum and derived nanoformulations are extremely superficial, i.e., it does not describe the molecular targets or elucidate the molecular mechanisms, nor does it highlight the most likely compounds underlying such effects. The authors solely refer to general anticancer, anti-inflammatory, or immunostimulating effects, such as pro-apoptotic effects and interference with angiogenesis, without detailing the cancer types and/or in vitro models of disease. (e.g., lines 393-409; 499-512; 593-610).

3. There are several scientific inaccuracies throughout the manuscript, such as in:

Lines 201-202: The assessment of “bioactivity” is not a parameter for the taxonomic identification or authentication of G. lucidum based products.

Lines 207: As a fungi (Basidiomycota) G. lucidum is not categorized as an “herb”.

Line 231: Simply mentioning "guanine" and "adenine" refers to the nitrogenous bases, not nucleotides. Furthermore, these are structural biomolecules and not actually categorized as being bioactive.

Lines 287-288: Ergosterol is a common component of fungal cell membranes that is already acknowledged as being devoid of any therapeutic potential.

Line 292: Ganodermic acids and ganodermanontriol are structurally classified as triterpenoids and should be referred to in the appropriate section of the manuscript.

Lines 363-364: While obtaining compounds with high purity might be limiting, extraction and purification methods do not impact the activity of single, purified compounds. A possible explanation would be different ionization states.

B) The current manuscript lacks scientific novelty, as there are several recent reviews covering the same content and more effectively fulfilling the same aim (doi: 10.1002/ptr.7215; 10.3389/fphar.2022.952027; 10.1186/s12935-023-03146-8; 10.1016/j.ijbiomac.2023.125199 ; 10.1039/d2fo01683d).

C) The manuscript requires moderate revision of the English writing style, which is frequently rudimentary and contains several typos. A few examples are highlighted below:

Lines 47-49; 139-144: The writing style is superficial and rudimentary.

Line 97: Revise “preclinial” to “preclinical”.

Line 115: Revise “delevery” to “delivery”.

Line 116: Revise “addpted” to “adopted”.

D) The authors should cite the sources of the pictures included in the figures, as they are not original.

For example, in Figure 1, the image of Ganoderma lucidum was obtained from online vendors and has been manipulated; the fungus appears elongated rather than rounded. Similarly, the image of Ganoderma sinense was obtained from an Amazon link.

E) Table 5 lacks data and the one from pages 41 to 42 is not numbered nor include a description.

Comments on the Quality of English Language

The manuscript requires moderate revision of the English writing style, which is frequently rudimentary and contains several typos. A few examples are highlighted below:

Lines 47-49; 139-144: The writing style is superficial and rudimentary.

Line 97: Revise “preclinial” to “preclinical”.

Line 115: Revise “delevery” to “delivery”.

Line 116: Revise “addpted” to “adopted”.